# BLADE: Block-Sparse Attention Meets Step Distillation for Efficient Video Generation

**Youping Gu**[1*]  **Xiaolong Li**[1*]  **Yuhao Hu**[2]  **Minqi Chen**[2]  **Bohan Zhuang**[1†]

[1]Zhejiang University  [2]Central Media Technology Institute, Huawei Technologies

[1]{youpgu71,xiaolong.ziplab,bohan.zhuang}@gmail.com
[2]huyuhao1@h-partners.com,chenminqi@huawei.com

## ABSTRACT

Diffusion transformers currently lead the field in high-quality video generation, but their slow iterative denoising process and prohibitive quadratic attention costs for long sequences create significant inference bottlenecks. While both step distillation and sparse attention mechanisms have shown promise as independent acceleration strategies, effectively combining these approaches presents critical challenges—training-free integration yields suboptimal results, while separately training sparse attention after step distillation requires prohibitively expensive high-quality video data. To overcome these limitations, we propose *BLADE*, an innovative data-free joint training framework that introduces: (1) an Adaptive Block-Sparse Attention (ASA) mechanism for dynamically generating content-aware sparsity masks to focus computation on salient spatiotemporal features, and (2) a sparsity-aware step distillation paradigm, built upon Trajectory Distribution Matching (TDM), directly incorporates sparsity into the distillation process rather than treating it as a separate compression step and features fast convergence. We validate BLADE on text-to-video models like CogVideoX-5B and Wan2.1-1.3B, and our framework demonstrates remarkable efficiency gains across different scales. On Wan2.1-1.3B, BLADE achieves a $14.10\times$ end-to-end inference acceleration over a 50-step baseline, and an $8.89\times$ speedup on the short-sequence model CogVideoX-5B. Crucially, the acceleration is achieved while maintaining generation quality comparable to the original 50-step baseline. On the VBench-2.0 benchmark, BLADE boosts the score of CogVideoX-5B to 0.569 (from 0.534) and Wan2.1-1.3B to 0.570 (from 0.563), results that are further corroborated by superior ratings in human evaluations. The Project is available at http://ziplab.co/BLADE-Homepage/.

## 1 INTRODUCTION

Diffusion models have emerged as the state-of-the-art for a wide array of generative tasks (Dhariwal & Nichol, 2021), achieving unprecedented quality in image synthesis (Cao et al., 2024; Esser et al., 2024; Labs et al., 2025) and now pushing the frontier in the complex domain of video generation (Blattmann et al., 2023; Xing et al., 2024). By modeling generation as a gradual reversal of a noising process (Ho et al., 2020), these models can produce diverse and high-fidelity content. However, for diffusion transformers, this power comes at a severe computational cost (Shen et al., 2025). The introduction of the temporal dimension dramatically inflates the complexity of the attention mechanism, which scales quadratically with sequence length (Wan et al., 2025; Yang et al., 2024; Kong et al., 2025). This, combined with the iterative nature of the denoising process, results in prohibitively slow inference speeds that hinder practical deployment.

To mitigate this critical efficiency bottleneck, two primary research directions have gained prominence: reducing the number of inference steps via step distillation (Song et al., 2023; Salimans & Ho,

---

[*]Equal contribution.
[†]Corresponding author.

2022; Liu et al., 2024; Zheng et al., 2024; Gu et al., 2023; Goodfellow et al., 2014; Yin et al., 2024) and lowering the per-step cost via sparse attention (Zhang et al., 2025b; Yuan et al., 2024; Zhang et al., 2025a; Li et al., 2025; Xu et al., 2025; Dao et al., 2022). However, effectively integrating these two powerful paradigms is non-trivial and presents a critical dilemma. A naive, training-free combination, where a pre-trained sparse attention mechanism is applied to a distilled model, yields suboptimal results because the distillation process is agnostic to sparse attention. Conversely, a sequential training pipeline that involves first performing step distillation and then fine-tuning the model for sparsity is equally impractical, as it re-introduces the need for prohibitively large and expensive high-quality video datasets, counteracting the key benefits of modern data-free distillation methods (Gu et al., 2023; Sauer et al., 2024; Luo et al., 2025).

The challenge of designing an appropriate sparse attention mechanism is further exacerbated in the video domain. Many existing methods rely on static, content-agnostic sparsity patterns (Zhang et al., 2025b; Li et al., 2025; Xi et al., 2025). These fixed patterns, such as rigid local windows or predetermined striding, fail to adapt to the dynamic and diverse spatiotemporal features of video content. Consequently, they often struggle to preserve important details and long-range dependencies, leading to significant quality degradation, especially at higher sparsity levels required for meaningful acceleration. In contrast, another line of work explores dynamically generated attention masks, which allow the sparsity pattern to adapt to content-specific structure. While dynamic masking methods such as VSA (Zhang et al., 2025c) improve the trade-off between efficiency and fidelity, they conceptually operate on structured 3D token grids. For irregular latent shapes, this design typically necessitates padding dimensions to align with tile boundaries, introducing computational overhead that can diminish practical sparsity gains. On the other hand, SpargeAttention (Zhang et al., 2025a) supports training-free inference but cannot be trained and exhibits limited sparsity. Limited flexibility and applicability restrict the widespread adoption of dynamic sparse attention in video generation.

This landscape highlights a clear need for a sparse attention mechanism that is computationally efficient, dynamically content-aware, and flexible enough to support arbitrary resolutions and both training-free and training-aware modes at high sparsity without sacrificing visual fidelity. To this end, we introduce *ASA*, a training-free sparse attention framework with dynamic token selection, capable of adapting to input content while maintaining high generation quality across various settings. For cases where training is permitted, we further present *ASA_G*, a distillation-based variant that leverages global token prediction to enable end-to-end training. Together, ASA and ASA_G offer a unified solution to both inference and training scenarios in efficient video generation.

Overall, this paper argues that a truly effective solution requires moving beyond treating distillation and sparsity as separate, post-hoc optimizations. We introduce *BLADE* (BLock-sparse Attention Meets step Distillation for Efficient video generation), a novel framework that pioneers the *synergistic, data-free joint training* of dynamic sparsity and step distillation. Our approach directly incorporates sparsity-awareness into the distillation process, allowing the student model to learn a compact and efficient trajectory from the teacher, conditioned on a dynamic attention pattern.

The main contributions of this work are as follows:

1. We propose *BLADE*, a *data-free joint training framework* that synergistically integrates an adaptive sparse attention mechanism directly into a sparsity-aware step distillation process, overcoming the limitations of prior sequential or training-free integration approaches.
2. We introduce Adaptive Block-Sparse Attention (ASA), a dynamic, content-aware, and hardware-friendly attention mechanism that generates sparsity masks on-the-fly to focus computation on salient features.
3. We demonstrate significant end-to-end inference acceleration on diverse models, achieving a **14.10×** speedup on Wan2.1-1.3B and a robust **8.89×** on the shorter-sequence CogVideoX-5B. Crucially, this acceleration is accompanied by a consistent *quality improvement*, with VBench-2.0 scores increasing for both Wan2.1-1.3B (0.563 → 0.570) and CogVideoX-5B (0.534 → 0.569).

## 2 RELATED WORK

### 2.1 VIDEO GENERATION WITH DIFFUSION MODELS

Recent years have witnessed remarkable progress in video generation, largely driven by the success of diffusion models (Ho et al., 2020; Song et al., 2021; Ma et al., 2025; Cao et al., 2024; He et al., 2023). These models have become the de facto standard for synthesizing high-fidelity and temporally coherent video content, achieving state-of-the-art results on various benchmarks (Huang et al., 2024; Zheng et al., 2025).

The operating principle of diffusion models is to learn the reversal of a fixed data corruption process. Specifically, a noisy sample $\mathbf{x}_t$ is generated by corrupting a clean sample $\mathbf{x}_0 \sim p_{\text{real}}$ using a simple formulation: $\mathbf{x}_t = \alpha_t \mathbf{x}_0 + \sigma_t \boldsymbol{\epsilon}$, where $\boldsymbol{\epsilon} \sim \mathcal{N}(\mathbf{0}, \mathbf{I})$ is standard Gaussian noise. The positive scalars $\alpha_t$ and $\sigma_t$ are dictated by a noise schedule, which controls the signal-to-noise ratio at each timestep $t$ (Karras et al., 2022).

The model's task is to learn this reversal. A network, often termed a denoiser $\boldsymbol{\mu}_\theta(\mathbf{x}_t, t)$, is trained to predict the original clean sample $\mathbf{x}_0$ from its corrupted version $\mathbf{x}_t$. This learned denoiser provides an estimate of the score function (Song et al., 2021):

$$s_\theta(\mathbf{x}_t, t) = \nabla_{\mathbf{x}_t} \log p_{\text{real},t}(\mathbf{x}_t) \approx -\frac{\mathbf{x}_t - \alpha_t \boldsymbol{\mu}_\theta(\mathbf{x}_t, t)}{\sigma_t^2}. \tag{1}$$

Generation is then achieved by starting with pure noise $\mathbf{x}_T \sim \mathcal{N}(\mathbf{0}, \mathbf{I})$ and iteratively applying the learned denoising function to reverse the corruption process, step-by-step, until a clean sample $\mathbf{x}_0$ is obtained.

### 2.2 ACCELERATION VIA STEP DISTILLATION

Step distillation has emerged as a primary strategy for accelerating diffusion models (Song et al., 2023; Salimans & Ho, 2022; Liu et al., 2024; Zheng et al., 2024; Gu et al., 2023; Goodfellow et al., 2014). The goal is to transfer the knowledge from a slow "teacher" model (e.g., a 50-step sampler) to a faster "student" model that can generate comparable results in very few steps (e.g., 1–8 steps). Early methods like Progressive Distillation (Salimans & Ho, 2022; Luhman & Luhman, 2021) iteratively halve the number of sampling steps. Distillation strategies can be broadly categorized into output distillation, which trains the student to match the final output of a multi-step teacher process, and trajectory distillation (Luhman & Luhman, 2021; Song et al., 2023), which guides the student to follow the teacher's intermediate generation path. Trajectory Distribution Matching (TDM) represents a recent and sophisticated advancement in this area (Luo et al., 2025). TDM unifies the concepts of distribution matching and trajectory matching. Instead of enforcing a strict instance-level match of the trajectory, it aligns the *distribution* of the student's intermediate samples with the teacher's corresponding diffused distributions at each step. A key advantage of TDM is that it is a *data-free* method; it does not require access to the original, often proprietary, training dataset, relying only on the pre-trained teacher model to generate guidance signals. This makes it a highly practical and versatile distillation framework, which we adopt as the foundation for our work.

### 2.3 VIDEO-SPECIFIC SPARSE ATTENTION

Several promising approaches have been proposed to accelerate attention computation, each with distinct mechanisms and trade-offs. Early methods such as STA (Zhang et al., 2025b) and Radial Attention (Li et al., 2025) primarily utilize static attention masks. STA employs a fixed local window, a design choice that makes it most effective for specific input dimensions, while Radial Attention proposes a heuristic whose resulting sparsity is less pronounced on shorter sequences, limiting its adaptability. To introduce more dynamism, SVG (Xi et al., 2025) selects between two pre-defined masks, a binary choice that offers limited granularity and may create a trade-off between quality and sparsity. Other methods like SpargeAttention (Zhang et al., 2025a) also shows potential in training-free scenarios. However, it is not applicable to training, and its sparsity level must be kept moderately low to preserve video quality. VSA (Zhang et al., 2025c) introduces training and offers finer-grained control via fixed attention cubes, a design that influences the range of applicable resolutions. To bridge these varied trade-offs, we propose Adaptive Block-Sparse Attention (ASA),

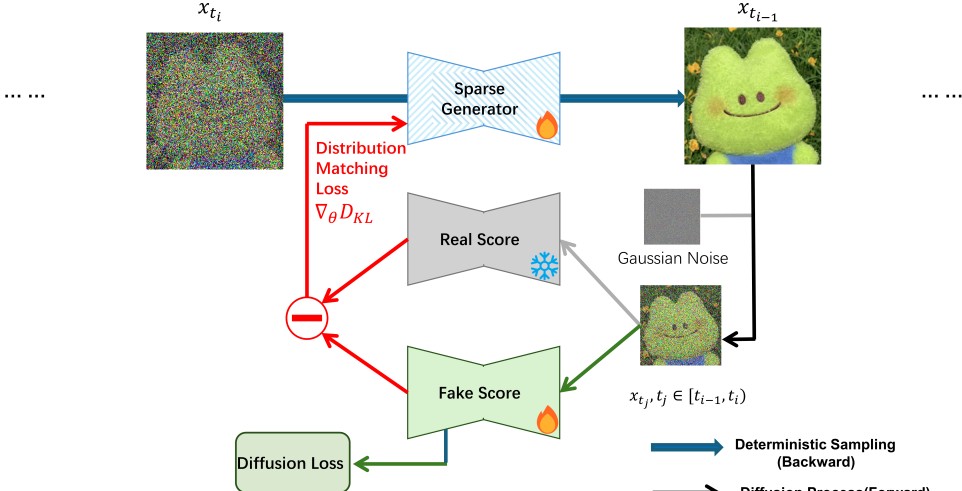

Figure 1: The training mechanism of BLADE within a single distillation interval $[t_{i-1}, t_i)$. The Sparse Generator $(G_\theta)$ denoises the input $\mathbf{x}_{t_i}$ to produce the sample $\mathbf{x}_{t_{i-1}}$. Crucially, this output is then re-corrupted with Gaussian noise to create an intermediate sample $\mathbf{x}_{t_j}$. A dedicated Fake Score model evaluates this re-noised sample. Its output is contrasted with the score from the Real Score model (which is the pre-trained teacher model) to compute the Distribution Matching Loss $(\nabla_\theta D_{KL})$. This loss directly updates the student generator, forcing it to align its generation trajectory with the teacher's at a distributional level.

a dynamic, content-aware mechanism that generates hardware-friendly sparsity masks on-the-fly, providing a unified solution for both training-free and distillation-based scenarios.

## 3 METHOD

### 3.1 OVERALL ARCHITECTURE

BLADE is a holistic framework for accelerating video diffusion models by synergistically integrating dynamic sparsity into a powerful step distillation process. As illustrated in Figure 1, our architecture is based on a student-teacher paradigm. The teacher, $f_\phi$, is a pre-trained, high-quality but computationally expensive multi-step diffusion model. The student, $G_\theta$, initially shares the same Transformer-based (DiT) (Peebles & Xie, 2023) architecture and weights as the teacher. Our key innovation, designed to enable few-step generation, is the replacement of the standard self-attention layers within the student with our proposed Adaptive Block-Sparse Attention (ASA) mechanism. The training process follows the Trajectory Distribution Matching (TDM) (Luo et al., 2025) paradigm. In each iteration, the sparse student model $G_\theta$ generates an intermediate trajectory. This trajectory is then guided to match the distribution of the teacher's trajectory via a data-free score distillation loss. This ensures the student learns to produce high-quality outputs while operating under the computational constraints imposed by ASA.

### 3.2 PRELIMINARIES: TRAJECTORY DISTRIBUTION MATCHING (TDM)

Trajectory Distribution Matching (TDM) (Luo et al., 2025) is an advanced distillation framework designed to create efficient, few-step diffusion models. Its core idea is to align the entire generation trajectory of a student model with that of a teacher model at the distribution level, rather than requiring an exact instance-level match. This is operationalized through a data-free score distillation process that relies on three key components:

1. The pre-trained teacher model $f_\phi$, which provides the real data score $s_\phi$.
2. The student generator $G_\theta$, which generates samples through $K$ denoising steps defined by the time sequence $\{t_i\}_{i=0}^K$, where $0 = t_0 < t_1 < \cdots < t_K = T$.

3. A fake score model $f_\psi$, which is trained to approximate the student's intractable sample score and provides the estimated score $s_\psi$.

The training process involves two intertwined objectives, one for the fake score model and one for the student generator.

**Training the fake score model** $(f_\psi)$.  The score distillation process requires the student model's score function $\nabla_{\mathbf{x}} \log p_{\theta,t}(\mathbf{x})$, which is intractable. TDM resolves this by introducing a fake score model $f_\psi$, a neural network trained concurrently to approximate the student's score. To ensure this approximation is accurate, the fake score model is parameterized as a denoiser and trained using the following objective:

$$\mathcal{L}(\psi) = \sum_{i=0}^{K-1} \mathbb{E}_{\mathbf{x}_{t_i} \sim p_{\theta,t_i}} \mathbb{E}_{\mathbf{x}_j \sim q(\mathbf{x}_j | \mathbf{x}_{t_i})} \| f_\psi(\mathbf{x}_j, j) - \mathbf{x}_{t_i} \|_2^2, \tag{2}$$

where the student-generated sample $\mathbf{x}_{t_i}$ at timestep $t_i$ serves as the denoising target. A noisy sample $\mathbf{x}_j$ is then created by perturbing $\mathbf{x}_{t_i}$ to an intermediate timestep $j$, and the model learns to recover $\mathbf{x}_{t_i}$ from $\mathbf{x}_j$.

**Training the student generator** $(G_\theta)$.  With access to both the teacher's score $s_\phi$ and the student's own score estimate $s_\psi$ (derived from the denoiser output $f_\psi$), the student generator $G_\theta$ can be trained. The objective is to minimize the KL divergence between the student's trajectory distribution and the teacher's trajectory distribution. This alignment is performed across the $K$ sampling intervals. The core distillation loss is:

$$\mathcal{L}(\theta) = \sum_{i=0}^{K-1} \lambda_i D_{\mathrm{KL}} \left( p_{\theta,t_i}(\mathbf{x}) \| p_{\phi,t_i}(\mathbf{x}) \right). \tag{3}$$

In practice, minimizing this KL divergence is achieved by matching the scores. The gradient of this objective is computed by replacing the student's intractable true score with the estimated score $s_\psi$. This results in the following gradient approximation:

$$\nabla_\theta \mathcal{L}(\theta) = \sum_{i=0}^{K-1} \sum_{j=t_i}^{t_{i+1}} \lambda_j [\nabla_{\mathbf{x}_j} \log p_{\theta,j}(\mathbf{x}_j) - s_\phi(\mathbf{x}_j, j)] \frac{\partial \mathbf{x}_{t_i}}{\partial \theta} \tag{4}$$

$$\approx \sum_{i=0}^{K-1} \sum_{j=t_i}^{t_{i+1}} \lambda_j [s_\psi(\mathbf{x}_j, j) - s_\phi(\mathbf{x}_j, j)] \frac{\partial \mathbf{x}_{t_i}}{\partial \theta}.$$

Following the TDM framework (Luo et al., 2025), this process is made both practical and memory-efficient through two key implementation choices. First, we ensure the distillation intervals $[t_i, t_{i+1}]$ are non-overlapping. This design allows a *single fake score model $f_\psi$* to be sufficient for all stages. Second, to conserve GPU memory, backpropagation through the student generator is constrained to only *one ODE step* at a time.

### 3.3 ADAPTIVE BLOCK-SPARSE ATTENTION (ASA)

A core design of our work is the Adaptive Block-Sparse Attention (ASA) mechanism, developed upon block-sparse attention. ASA leverages the prior that neighboring tokens in the latent representations of video often share similar semantics, which makes it reasonable for queries within the same block to share a mask, and pooling operation can keep meaningful semantic information. By allowing each query in a block to selectively attend to only the most relevant keys and values, ASA achieves superior performance over traditional static masks. In the following, we provide a detailed introduction to our method.

**Preprocessing: Locality-preserving token rearrangement.** The input matrix $Q$, $K$, and $V$, representing a flattened sequence of video tokens, are first restructured into blocks. A critical preliminary step is rearranging the tokens to preserve their inherent spatial locality, which is often disrupted by standard raster-scan tokenization. To this end, we employ a Gilbert space-filling curve (Zhang et al.,

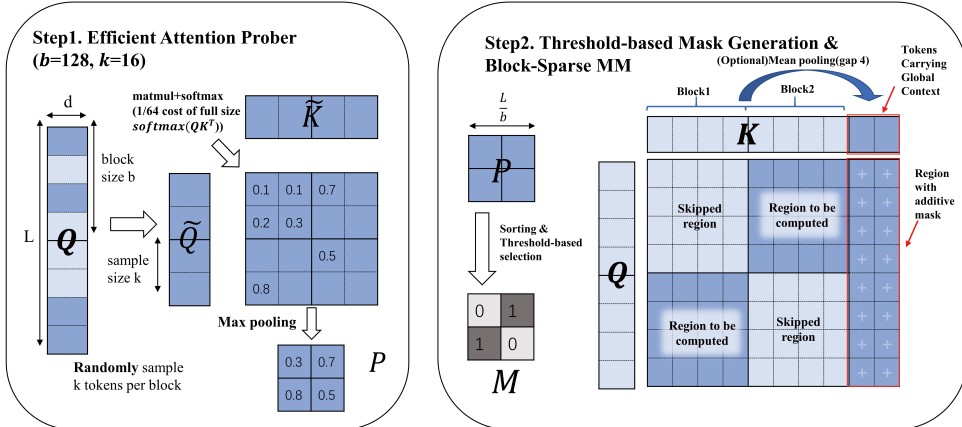

Figure 2: The two-stage process for Adaptive Block-Sparse Attention mask generation. (1) The *Efficient Attention Prober* samples a few representative tokens (e.g., $k = 16$) from each block to compute a low-cost max-pooled attention matrix $P$. (2) The *Threshold-based Mask Generator* sorts the scores in $P$ and selects the top blocks that contain a specified threshold (e.g., 95%), producing the final binary mask $M$. To enrich the context for training, we augment the key matrix $K$ by concatenating it with a pooled version: $K = \text{Concat}(K, \text{MeanPool}_n(K))$, where $\text{MeanPool}_n(K)$ denotes mean pooling over a window of size $n$. During attention computation, the original $K$ region uses the binary block mask $M$, while the pooled region receives a fixed additive mask of $\ln n$, softly guiding attention without disrupting sparsity.

2025a) to reorder the tokens before blocking. This ensures that the resulting blocks are more semantically coherent, containing spatially contiguous information, which significantly enhances the effectiveness of the subsequent threshold-based pruning.

**Step 1: Efficient block importance estimation.** Conceptually, one could compute the full, dense attention matrix $P = \text{softmax}(QK^\top/\sqrt{d_k})$, partition it into blocks of size $b \times b$, and then apply max-pooling over each block. This would yield a downsampled importance matrix, $P_{\text{imp}}$, where each element signifies the importance of the corresponding block. A sparse mask could then be generated by applying a threshold to each row of $P_{\text{imp}}$, allowing each query block to focus only on the most salient key-value blocks. However, the initial computation of the full matrix $P$ makes this method impractical for achieving actual acceleration.

To overcome this limitation, we propose an efficient online approximation. Instead of the full matrix, we sample $k$ representative tokens ($k < b$) from each block of $Q$ and $K$ to form smaller matrices, $Q_s$ and $K_s$. We then compute a much smaller, low-resolution attention map, $P_{\text{approx}}$, from these sampled tokens. The block importance matrix $P_{\text{imp}}$ is derived from this approximate map. This approach reduces the complexity of mask generation from $\mathcal{O}(N^2)$ to approximately $\mathcal{O}(N^2 \cdot (k/b)^2)$, where $N$ is the sequence length. This makes online mask generation feasible. This sampling-based scheme not only reduces the complexity of mask generation, but also improves the accuracy of block importance estimation. Unlike SpargeAttention (Zhang et al., 2025a), which collapses each block into a single mean token and derives importance from a coarse $N/b \times N/b$ attention map, ASA retains intra-block structure by computing attention over sampled tokens and then applying max-pooling within each sub-block. This finer-grained approximation enables ASA to better capture salient patterns within each block. Detailed experimental comparisons with SpargeAttention are provided in Table 8.

**Step 2.1: Sparse mask construction.** Once the block importance matrix $P_{\text{imp}}$ is obtained, we generate the final sparse attention mask based on a threshold-based masking strategy. Specifically, we sort each row of $P_{\text{imp}}$ in descending order and include the minimal number of key blocks such that their cumulative attention scores exceed a specified threshold (e.g., 90%). This threshold-based dynamic pruning preserves the most salient attention paths while skipping less informative blocks, offering a flexible trade-off between accuracy and efficiency.

Table 1: Video Quality Evaluation on VBench-2.0.

| Model | Method | Sparsity | Total | Creativity | Commonsense | Controllability | Human | Physics | Speedup |
|-------|--------|----------|-------|------------|-------------|-----------------|-------|---------|---------|
| CogvideoX-5B | Baseline | - | 0.534 | 0.458 | **0.523** | 0.341 | 0.808 | 0.539 | 1× |
| | FA2 | - | 0.539 | 0.458 | 0.498 | 0.354 | **0.813** | 0.570 | 7.93× |
| | *ASA_G* | 0.82 | **0.569** | **0.546** | 0.514 | **0.367** | 0.802 | **0.618** | **8.89×** |
| Wan2.1-1.3B | Baseline | - | 0.563 | 0.508 | **0.549** | **0.338** | 0.820 | 0.600 | 1× |
| | FA2 | - | **0.580** | **0.631** | 0.485 | 0.311 | 0.841 | **0.631** | 9.37× |
| | STA | 0.74 | 0.528 | 0.504 | 0.471 | 0.265 | 0.855 | 0.543 | 10.53× |
| | *ASA_G* | 0.8 | 0.570 | 0.472 | 0.532 | 0.312 | **0.918** | 0.617 | **14.10×** |

*Note:* **Baseline refers to the official 50 steps baseline. All methods except the Baseline are distilled to 8 steps using TDM.**

The resulting binary mask is then used to restrict the computation of attention during both training and inference, ensuring that the majority of computational resources are focused on the most relevant interactions. We provide the pseudocode of ASA in Algorithm 3.

**Step 2.2: Computation.** Based on this mask generation technique, we introduce two variants of our mechanism tailored to different application scenarios:

*1) Standard ASA (Training-Free):* In its primary form, the generated binary sparse mask $M$ is directly integrated with a block-sparse attention kernel. This variant can be applied to pre-trained models without any retraining, offering a direct inference speed-up by focusing computation on fine-grained, salient information.

*2) ASA with Global Tokens (for Distillation):* To mitigate potential global information loss at high sparsity ratios, we introduce an enhanced variant. We augment the Key ($K$) and Value ($V$) by creating a set of *"global tokens"*. These are generated by applying mean pooling over a window of size $n$, reducing the sequence length to $1/n$ of the original lengths of $K$ and $V$. The augmented $K$ are formed as $K_{\text{aug}} = \text{Concat}(K, \text{MeanPool}_n(K))$ (and similarly for $V$). During attention computation, a query's interaction with the original $K$ region is governed by the binary sparse mask $M$, preserving fine-grained details. For the augmented "global tokens" region, we apply a fixed additive mask of $\ln(n)$ to the pre-softmax scores. This bias compensates for the averaging effect of mean pooling, ensuring that each global token contributes attention as if it represents the full importance of its $n$ constituent fine-grained tokens. This softly guides every query to maintain awareness of the global context, preventing catastrophic information loss when most blocks are pruned.

Throughout this paper, we refer to the standard implementation as **ASA** and the augmented version as **ASA with Global Tokens (ASA_G in short)**.

### 3.4 SPARSITY-AWARE DISTILLATION

A cornerstone of the BLADE framework is the principle of sparsity-aware distillation. Unlike previous approaches that apply sparsity as a post-training compression step, we integrate ASA directly into the TDM training loop. At every training iteration, the student model $G_\theta$ generates its trajectory *using the ASA mechanism*. The distribution matching loss then updates the student's weights to improve its output quality *given these dynamic sparsity constraints*. This co-design strongly regularizes the model, forcing it to learn a robust, semantic representation that often yields superior perceptual quality.

## 4 EXPERIMENT

### 4.1 EXPERIMENTAL SETUP

**Models.** We evaluate BLADE on two text-to-video diffusion models: CogVideoX-5B (Hong et al., 2022) and Wan2.1-1.3B (Wan et al., 2025). These models represent different architectures and scales, allowing us to test the generalizability of our approach.

**Dataset.** Our training process is guided by a dataset of 10,000 text prompts. These prompts were sampled from the JourneyDB benchmark (Sun et al., 2023) and subsequently enhanced for quality and diversity using the Qwen2.5-3B-Instruct (Team, 2024) model.

**Implementation details.** Unless otherwise specified, we use a block size $b = 128$, $k = 16$ sampled tokens per block for the attention prober. Distillation is typically run for 100-200 iterations. Experiments on CogVideoX-5B and Wan2.1-1.3B were conducted on a cluster of 8 A800(80GB) GPUs. We use a suite of standard metrics to evaluate performance:VBench-1.0 (Huang et al., 2024), VBench-2.0 (Zheng et al., 2025), SSIM & PSNR (Hor & Ziou, 2010), Human Evaluation.

**Compared methods.** ASA_G, ASA, STA (Zhang et al., 2025b), and RaA (Li et al., 2025) respectively denote using our adaptive attention, its training-free variant, Sliding Tile Attention (Zhang et al., 2025b), and Radial Attention (Li et al., 2025). FA2 refers to FlashAttention-2 (Dao, 2024).

## 4.2 MAIN RESULTS: EFFICIENCY AND QUALITY

Our experiments demonstrate that Video-BLADE achieves significant acceleration without compromising, and often improving, generation quality.

**Quality Analysis.** Table 1 presents the VBench-2.0 benchmark results for CogVideoX-5B and Wan2.1-1.3B across several methods, including our proposed ASA_G, the sparse baseline STA, FA2, and the 50-step dense baseline.

For *CogVideoX-5B*, ASA_G delivers consistent and comprehensive improvements across all major quality dimensions. It achieves the highest overall VBench-2.0 score (0.569), outperforming both the 50-step baseline and FA2, and leads in Creativity, Controllability, and Physicsall key for generating plausible and engaging video content. Notably, ASA_G achieves this performance using only 8 decoding steps over a short 17k-token sequence, resulting in an **8.89× speedup** while simultaneously improving generation quality. These results demonstrate that even with extremely constrained sequence lengths, ASA_G achieves robust generation quality.

For *Wan2.1-1.3B*, ASA_G continues to show clear advantages. It achieves a strong VBench-2.0 score (0.570), the highest Human Fidelity (0.918), and strong Physics performance, all while operating at just **7.09%** of the original inference time (14.10 speedup). Compared to STA, which shares similar sparsity, ASA_G performs significantly better in almost all metrics. Although FA2 slightly outperforms ASA_G in total score, its performance on controllability is weaker and comes at a higher computational cost. A gallery-style visual comparison, showcasing video samples across diverse models and inference strategies, is presented in the Appendix F.

An intriguing observation from our results is that BLADE, despite its high sparsity and few inference steps, can surpass the quality of the 50-step dense baseline. We attribute this phenomenon to a regularization effect induced by our joint training framework. The long, iterative trajectory of the 50-step teacher can sometimes accumulate numerical errors or overfit to noisy, less coherent details. In contrast, our sparsity-aware distillation compels the student model to learn a more direct and stable generation path (a principle that echoes findings in prior works like DMD2 (Yin et al., 2024)), forcing it to capture the most essential semantics while implicitly filtering out the "detours" and noise from the teacher's process. The adaptive sparsity further aids this by focusing computation only on the most salient features. We provide a visual corroboration of this effect with attention map analyses in the Appendix B. The resulting model is therefore not merely a faster approximation but can be a more *robust and coherent generator*. We evaluate our models on VBench-2.0, which places greater emphasis on semantic faithfulnessassessing how well the generated videos preserve high-level meaning rather than just pixel-wise accuracy. This aligns closely with the strengths of our approach.

**Efficiency analysis.** At the kernel level, our ASA implementation achieves a **3.30×** speedup over the standard dense attention used in the 8-step FA2 baseline (22.21 ms vs. 73.25 ms), benefiting from an effective sparsity rate of 0.798. This low-level gain directly translates to a substantial end-to-end acceleration: our ASA-based model completes generation in **24.00 seconds**, compared to **36.11 seconds** for its dense counterpartyielding a **1.504×** E2E speedup.

Notably, while the kernel speedup is more than 3×, the E2E gain is sub-linear. This suggests that attention is no longer the dominant bottleneck in the distilled model; instead, other operations (e.g.,

Table 2: Efficiency analysis on Wan2.1-1.3B (test on an H20).

| Metric | FA2-50 | FA2-8 | *ASA-8* |
|---|---|---|---|
| Kernel Time (ms) | 73.25 | 73.25 | **22.21** |
| Kernel Speedup | 1.00× | 1.00× | **3.30×** |
| E2E Time (s) | 338.41 | 36.11 | **24.00** |
| E2E Speedup | 1.00× | 9.37× | **14.10×** |

*Note: The number suffix (e.g. FA2-50) indicates the number of inference steps used in each model.*

Table 3: Comparison of training-free sparse attention methods on Wan2.1-1.3B (8-step distilled model).

| Method | Sparsity | PSNR | SSIM |
|---|---|---|---|
| STA | 0.74 | 16.72 | 0.6190 |
| SVG | 0.75 | 16.68 | 0.6390 |
| *ASA* | 0.75 | **19.55** | **0.7433** |
| RaA | 0.50 | 22.07 | 0.8191 |
| *ASA* | 0.50 | **22.20** | **0.8290** |

the VAE encoder/decoder and non-attention layers within the transformer) begin to dominate the runtime. This shift validates the effectiveness of our targeted kernel optimization in minimizing attention overhead within modern diffusion pipelines.

### 4.3 COMPARISON OF SPARSE ATTENTION MECHANISMS

To isolate the performance of the ASA mechanism itself, we compare it against other sparse attention methods in a training-free inference setting on Wan2.1-1.3B. For sparse inference, the first two steps adopt FA2, while the remaining steps use sparse attention. Table 3 shows that at a similar sparsity level, ASA significantly outperforms STA (Zhang et al., 2025b), RaA (Li et al., 2025) and SVG (Xi et al., 2025) in both PSNR and SSIM, establishing its superiority as a dynamic attention mechanism. Videos sampled by different methods are shown in Figure 3. Further ablation studies, including human evaluation results, are provided in the Appendix A.

| FA2 | ASA (Ours) | STA | SVG |
|---|---|---|---|

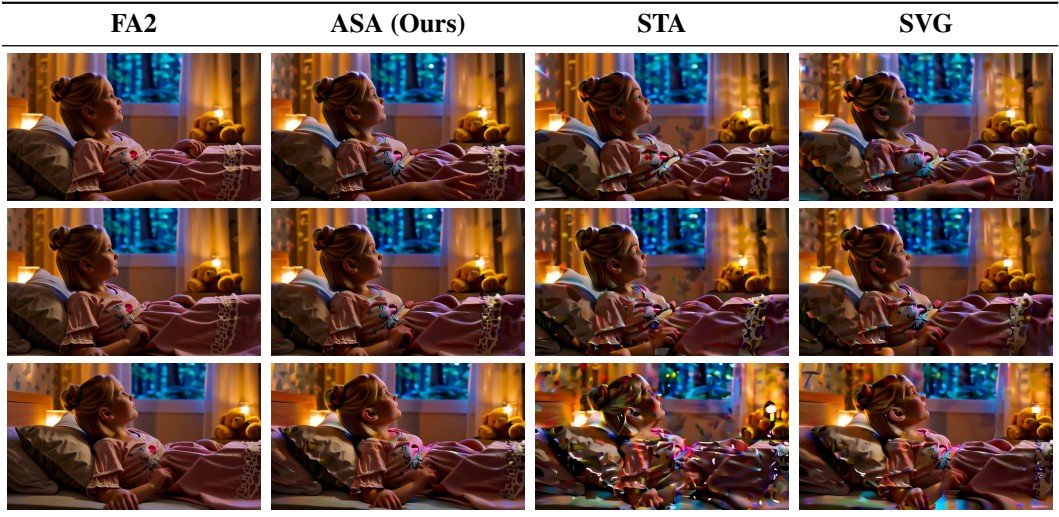

Figure 3: Comparison of generated videos at frame 0,40,80 for the prompt *"A tranquil tableau of bedroom"*. Each row shows the same frame index across 4 methods.

### 5 CONCLUSION AND FUTURE WORK

In this paper, we have presented BLADE, a novel framework that effectively addresses the critical efficiency challenge in video diffusion models. By synergistically co-designing a dynamic, content-aware adaptive block-sparse attention mechanism with a data-free trajectory distribution matching distillation process, our method achieves significant inference acceleration without sacrificing generation quality. Our results demonstrate that by making the model sparsity-aware during training, it often achieves superior visual quality and intrinsic faithfulness (Zheng et al., 2025) compared to both the original multi-step teacher and a densely distilled student model. Our contributions are

validated through extensive experiments on various video models, demonstrating marked improvements in kernel-level efficiency, end-to-end inference speed, and generation quality as measured by both automated benchmarks (VBench-2.0) and human evaluations.

**Limitations and future work.** While BLADE exhibits strong performance, we acknowledge several limitations that point to promising directions for future research. First, our current experiments are limited to video sequences of moderate length. Extending and validating the ASA mechanism for generating minute-long videos with hundreds of thousands of tokens remains an important next step. Additionally, our current ASA kernel is implemented in Triton for simplicity, which prevents it from fully realizing its theoretical speedup. Future work will focus on developing a more optimized CUDA implementation to better leverage the efficiency potential of ASA. These directions underscore the importance of evaluating ASA in more demanding settings and exploring further architectural enhancements. Lastly, the idea of sparsity-aware training as a form of regularization shows promise and could be extended to other generative domains beyond video synthesis, such as 3D content generation and high-resolution image synthesis.

## 6 ACKNOWLEDGEMENT

This work was supported by the Central Media Technology Institute, Huawei Technologies, through a technology cooperation.

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

# A ADDITIONAL EXPERIMENTS

We conducted a comprehensive set of experiments to analyze the contribution of each component and investigate the performance of BLADE under broader settings.

## A.1 IMPACT OF REARRANGEMENT STRATEGY

Table 4: Ablation results for the token rearrangement strategy, evaluated with the VBench-1.0 quality score.

| Configuration | Quality Score |
|---|---|
| Without Rearrange | 0.779 |
| With Rearrange (Ours) | **0.788** |

We validated the importance of the Gilbert rearrangement strategy. As shown in Table 4, CogVideoX-5B model distilled (using ASA) with this strategy achieve a higher VBench-1.0 quality score (0.788) compared to those without it (0.779), confirming its role in preserving spatial locality for more effective block-wise pruning.

## A.2 IMPACT OF ADDITIVE MASK AND GLOBAL TOKEN IN ASA_G

Table 5: Effect of Global Token (G) and Additive Mask (AM) in ASA on CogVideoX-5B (VBench-2.0).

| Config | Sparsity (%) | VBench-2.0 |
|---|---|---|
| ASA | 0.8 | 0.539 |
| ASA_G | 0.82 | **0.569** |
| ASA_G_w/o_AM | 0.82 | 0.559 |
| Baseline-50 | - | 0.534 |

**Note:** G = Global Token, AM = Additive Mask. Baseline-50 is the original 50-step FA2 model.

We conduct ablation studies on VBench-2.0 to validate the effectiveness of our key designs: the **Global Token (G)** and the **Additive Mask (AM)**. As shown in Table 5, our base model, ASA, already surpasses the baseline (0.539 vs. 0.534). Upon integrating the GT, the performance of our model, ASA_G, significantly leaps to **0.569**. This substantial gain underscores the critical role of GT in aggregating global spatio-temporal information. Furthermore, removing the AM from the full model (*i.e.*, ASA_G_w/o_AM) leads to a noticeable performance drop to 0.559, which confirms the necessity of AM in preserving model integrity under the sparse attention mechanism. Collectively, these results demonstrate that both GT and AM are indispensable components, synergistically contributing to the superior performance of our final model.

## A.3 IMPACT OF BLOCK SIZE CONFIGURATION

Table 6: Ablation study on block size configuration for ASA on Wan2.1-1.3B (sparsity ratio 0.8). Smaller block sizes offer finer granularity but incur higher overhead.

| Block Size (Q×K) | PSNR ↑ | SSIM ↑ | LPIPS ↓ |
|---|---|---|---|
| $64 \times 64$ | **22.24** | **0.818** | **0.144** |
| $128 \times 64$ | 22.05 | 0.803 | 0.162 |
| $128 \times 128$ | 21.75 | 0.793 | 0.169 |

Block size determines the granularity of the attention masking, acting as a pivotal hyperparameter in the ASA mechanism. We analyze three block configurations ($64 \times 64$, $128 \times 64$, and $128 \times 128$)

under a constant sparsity ratio of 0.8. Table 6 demonstrates a monotonic improvement in generation quality as block size decreases. Specifically, the $64 \times 64$ configuration outperforms the coarser $128 \times 128$ setting by 0.49 in PSNR and 0.025 in SSIM.

This performance gain stems from the finer granularity of smaller blocks, which allows the ASA mechanism to preserve salient regions with higher precision, adapting more effectively to local semantic structures. However, finer blocking introduces non-negligible computational overhead during mask generation and memory access. Consequently, we adopt $128 \times 128$ in our main experiments as a strategic trade-off, prioritizing system throughput while maintaining competitive generation quality.

## A.4  IMPACT OF ATTENTION THRESHOLD

Table 7: Ablation study on the attention threshold ($\tau$) in ASA. Experiments are conducted on Wan2.1-1.3B (50 steps; ASA enabled after a 12-step warm-up). $\tau$ governs the sparsity-quality frontier.

| Threshold ($\tau$) | Sparsity | PSNR ↑ | SSIM ↑ | LPIPS ↓ |
|---|---|---|---|---|
| 0.9 | 0.73 | **23.93** | **0.856** | **0.106** |
| 0.8 | 0.80 | 22.08 | 0.812 | 0.153 |
| 0.7 | 0.84 | 20.46 | 0.750 | 0.209 |
| 0.6 | 0.87 | 19.20 | 0.712 | 0.243 |
| 0.5 | 0.92 | 15.48 | 0.583 | 0.436 |

The threshold $\tau$ directly modulates the trade-off between computational sparsity and generation fidelity. By varying $\tau$ from 0.5 to 0.9, we observe in Table 7 that higher thresholds preserve more attention blocks, naturally leading to superior metrics. A threshold of $\tau = 0.9$ yields near-dense quality (PSNR 23.93) but results in a lower sparsity of 0.73. Conversely, aggressive pruning with $\tau = 0.5$ achieves high sparsity (0.92) but causes a structural collapse in quality (PSNR drops to 15.48), indicating that critical attention contexts are being discarded.

We identify $\tau = 0.8$ as the optimal operating point, achieving a sparsity of 0.80 without severe degradation in perceptual quality (SSIM > 0.8). This ablation further shows that ASA supports flexible deployment: higher thresholds are suitable for quality-critical applications, while lower thresholds benefit latency-sensitive scenarios where minor artifacts are acceptable. Visual examples across different sparsity levels are provided in Appendix F.1.

## A.5  SCALABILITY TO LARGER MODELS WITH LONGER SEQUENCES: WAN2.1-14B

Table 8: Comparison of training-free sparse attention methods on Wan2.1-14B. ASA achieves the best trade-off between sparsity and visual quality.

| Method | Sparsity | PSNR ↑ | SSIM ↑ | LPIPS ↓ |
|---|---|---|---|---|
| SVG | 0.75 | 24.86 | 0.823 | 0.094 |
| Sparge | 0.77 | 24.03 | 0.808 | 0.117 |
| STA | 0.75 | 25.00 | 0.845 | 0.079 |
| **ASA (Ours)** | 0.77 | **26.05** | **0.865** | **0.050** |

To validate the scalability of ASA to larger architectures and longer video sequences, we benchmark it against state-of-the-art training-free sparse attention methods on Wan2.1-14B. As shown in Table 8, ASA maintains superior performance at comparable sparsity levels ($\approx 0.75$–0.77). Notably, our method achieves an exceptionally low LPIPS of 0.050, outperforming the strongest baseline, STA, by a significant margin (0.029 improvement). Compared to SpargeAttention, ASA improves PSNR by over 2.0 and reduces perceptual loss by more than 50% (0.050 vs. 0.117), underscoring its capability to preserve fine-grained semantic details even when processing extended contexts. These results, consistent with the 1.3B evaluation, demonstrate that ASA's dynamic masking strategy is

robust across both model scales and sequence lengths, effectively mitigating the quality degradation observed in static pruning methods on large-scale, long-context foundation models.

## A.6 FEASIBILITY ANALYSIS AT LOW-STEP INFERENCE

To investigate the feasibility of BLADE under strict computational constraints, we extend the distillation process to a low-step regime of four sampling steps. Table 9 directly compares these 4-step sparse models against the standard 50-step dense baselines.

Notably, our method achieves substantial end-to-end inference speedups of $15.2\times$ on CogVideoX-5B and $17.6\times$ on Wan2.1-1.3B. Despite this aggressive reduction in sampling steps, both models maintain superior VBench Total scores compared to the 50-step baselines (CogVideoX: 0.562 vs. 0.534; Wan2.1: 0.570 vs. 0.563). These results validate that our joint training framework remains highly effective even in few-step scenarios, ensuring high-quality generation alongside extreme acceleration.

Table 9: VBench-2.0 comparison of 4-step BLADE models against 50-step baselines.

| Model | Method | Creativity | Commonsense | Controllability | Human | Physics | Total |
|---|---|---|---|---|---|---|---|
| CogVideoX-5B | Baseline | **0.458** | 0.523 | 0.341 | 0.808 | 0.539 | 0.534 |
| | ASA_G | 0.425 | **0.553** | **0.389** | **0.840** | **0.606** | **0.562** |
| Wan2.1-1.3B | Baseline | **0.508** | 0.549 | **0.338** | 0.820 | **0.600** | 0.563 |
| | ASA_G | 0.467 | **0.564** | 0.333 | **0.897** | 0.594 | **0.570** |

## A.7 HUMAN EVALUATION RESULTS

Table 10: Human preference: 8-step models vs. 50-step baseline.

| Comparison | Win | Lose | Tie |
|---|---|---|---|
| *CogVideoX-5B* | | | |
| ASA_G (Ours) vs. Baseline | 16 | 10 | 24 |
| *Wan2.1-1.3B* | | | |
| ASA_G (Ours) vs. Baseline | 10 | 12 | 28 |
| STA vs. Baseline | 0 | 26 | 24 |

We conducted a human preference study to evaluate our efficient 8-step ASA_G model(sparsity ratio 0.8) and 8-step STA model(sparsity ratio 0.74) against the standard 50-step baseline. The evaluation was performed using 50 diverse video prompts. The aggregated results are shown in Table 10.

For the CogVideoX-5B model, ASA_G was preferred or rated equally in 80% of comparisons, while achieving an **8.89×** speedup in inference time. For Wan2.1-1.3B, ASA_G achieved a 56% tie rate, yielding a 76% non-inferiority rate overall, while reducing inference time to just **7.09%** of the baseline. In contrast, STA was consistently outperformed by the baseline, with 0 wins and a 52% loss rate. These results highlight that ASA_G maintains high visual fidelity despite aggressive acceleration, validating its effectiveness for practical deployment.

## B MASK VISUALIZATION

To elucidate the mechanism by which our proposed sparse attention method achieves both significant acceleration and enhanced quality in video generation, we conducted a visualization analysis of the model's internal attention patterns. We hypothesize that constraining the model to operate within a limited computational budget compels it to disregard low-information, redundant areas (e.g., static backgrounds) and instead concentrate its focus more efficiently on core semantic objects within the scene.

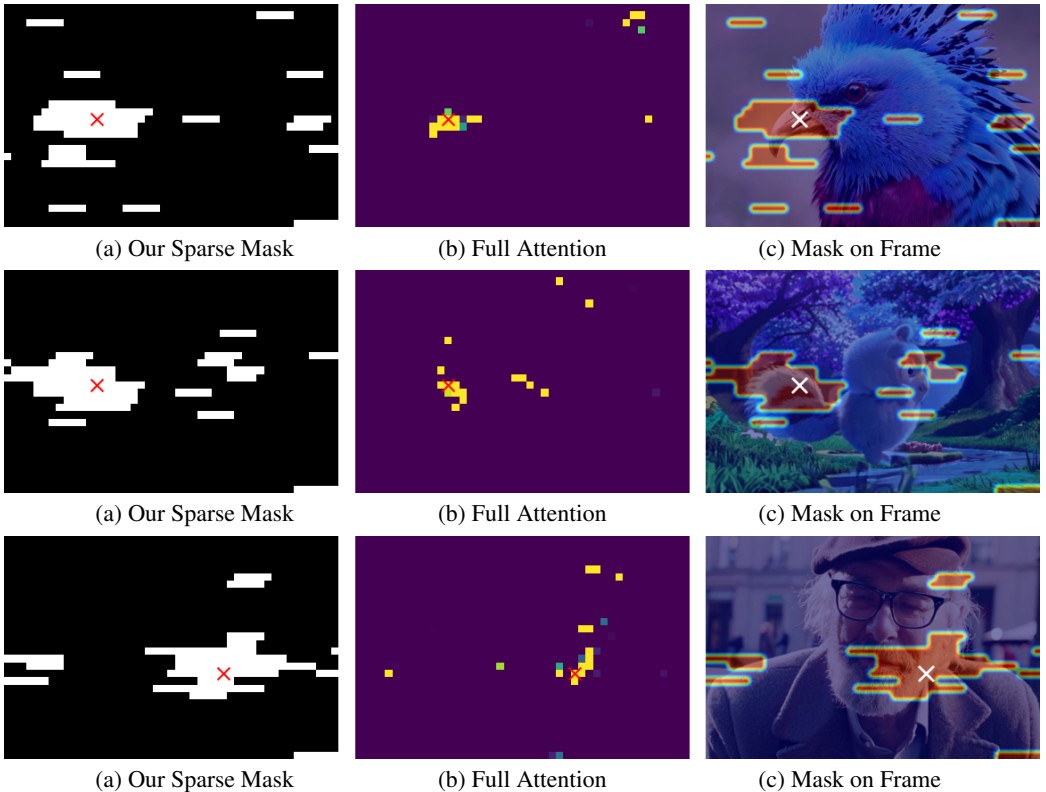

Figure 4: Visualization and analysis of attention masks. Our sparse method (a) is shown to capture the most salient regions identified by full attention (b), effectively focusing on key semantic objects within the frame (c).

Figure 4 offers a direct and intuitive validation of this hypothesis. Each row illustrates the attention behavior for a single sample, comparing our method against a standard full attention baseline across diverse scenes. Specifically, the three sub-figures within each composite image correspond to:

(a) Our Sparse Mask: This visualizes the attention mask produced by our sparse method for a single query patch Q. The white areas denote the spatial positions of key patches K that are retained for the attention score calculation. Conversely, the extensive black regions are the positions pruned by our method, where attention is not computed, effectively masking out non-salient information prior to the softmax operation.

(b) Full Attention Map: As a baseline, this map displays the raw attention weight distribution for the same query patch without sparsity constraints. Brighter colors (e.g., yellow) indicate higher attention scores.

(c) Mask Overlaid on Frame: The sparse mask, highlighted as a semi-transparent red overlay, is superimposed on the actual video frame to intuitively show the spatial locus of attention.

Across the three distinct scenes (a bird, a cartoon, and an elderly man), it is evident that although our method prunes a substantial number of computations (as shown in column a), the retained attention regions precisely cover the core semantic objects, such as the bird's beak, the cartoon character's tail, and the man's beard.Notably, the regions selected by our sparse mask exhibit a high degree of overlap with the highest-scoring areas in the full attention map. This provides strong evidence that our sparsity strategy effectively identifies and preserves the most salient semantic information while filtering out redundant background noise. This focusing mechanism offers a plausible explanation for the unexpected improvement in the model's generation quality.

## C PSEUDOCODE

Inspired by the pseudocode in SeerAttention, we adapt it to our implementation to get the max-pooling of attention map $P$ from the downsampled $Q$, $K$ input. The process is detailed in Algorithms 2 and 1.The pseudocode of ASA is detailed in 3.

---

**Algorithm 1** GetMaxPooledAttnMap

---

**Input:** $Q, K \in \mathbb{R}^{H \times \hat{S} \times d}$, pooling size $\hat{b}$, scale factor $s$
**Output:** $A \in \mathbb{R}^{H \times T_r \times T_r}$, $T_r = \lceil \hat{S}/\hat{b} \rceil$
 1: Initialize $M$ as $-\infty$ with shape $(H, \hat{S})$
 2: Initialize $\ell$ as 0 with shape $(H, \hat{S})$
 3: Initialize $R$ as $-\infty$ with shape $(H, \hat{S}, T_r)$
 4: **for** each head $h$ **do**
 5:     Split $Q_h$ $K_h$ into $T_r$ blocks: $Q_1, \ldots, Q_{T_r}$ $K_1, \ldots, K_{T_r}$
 6:     **for** $i \leftarrow 1$ to $T_r$ **do**
 7:         $\widetilde{M} \leftarrow M[h, (i-1) * \hat{b} : i * \hat{b}]$
 8:         $\widetilde{\ell} \leftarrow \ell[h, (i-1) * \hat{b} : i * \hat{b}]$
 9:         $\widetilde{R} \leftarrow R[h, (i-1) * \hat{b} : i * \hat{b}, :]$
10:         **for** $j \leftarrow 1$ to $T_r$ **do**
11:             $s_{ij} \leftarrow Q_i \cdot K_j^\top \cdot s$
12:             $m_{ij} \leftarrow \text{rowmax}(s_{ij})$, $\tilde{P}_{ij} \leftarrow \exp(s_{ij} - m_{ij})$
13:             $\tilde{\ell}_{ij} \leftarrow \text{rowsum}(\tilde{P}_{ij})$, $m_{\text{new}} \leftarrow \max(\widetilde{M}, m_{ij})$
14:             $\widetilde{\ell} \leftarrow e^{\widetilde{M} - m_{\text{new}}} \cdot \widetilde{\ell} + e^{m_{ij} - m_{\text{new}}} \cdot \tilde{\ell}_{ij}$
15:             $\widetilde{M} \leftarrow m_{\text{new}}$, $\widetilde{R}[:, j] \leftarrow m_{ij}$
16:         **end for**
17:         **for** $j \leftarrow 1$ to $T_r$ **do**
18:             $s_{ij} \leftarrow e^{\widetilde{R}[:,j] - \widetilde{M}}$, $s_{ij} \leftarrow s_{ij}/\widetilde{\ell}$
19:             $A[h, i, j] \leftarrow \max(s_{ij})$
20:         **end for**
21:     **end for**
22: **end for**

---

**Algorithm 2** Compute Block Importance Score

---

**Input:** Query $Q$, Key $K \in \mathbb{R}^{H \times S \times d}$, block size $b = 128$, tokens per block $k = 16$, scale factor $s$
**Output:** $P \in \mathbb{R}^{H \times T_r \times T_r}$ where $T_r = \lceil S/b \rceil$
 1: Make length divisible by $b$: $Q_p \leftarrow \text{Pad}(Q, b)$, $K_p \leftarrow \text{Pad}(K, b)$
 2: Sample $k$ tokens per block:
 3:     $\widetilde{Q} \leftarrow \text{BlockSample}(Q_p, b, k)$  $\widetilde{K} \leftarrow \text{BlockSample}(K_p, b, k)$
 4: $P \leftarrow \text{GetMaxPooledAttnMap}(\widetilde{Q}, \widetilde{K}, k, s)$

---

---

**Algorithm 3** ASA Mask Generation

---

**Require:** $Q, K \in \mathbb{R}^{N \times d}$, block size $b$, sample size $k$, threshold $\tau$
 1: Rearrange tokens using Gilbert curve
 2: Partition $Q, K$ into $N_b = N/b$ blocks
 3: Randomly sample $k$ tokens from each block to get $Q_s, K_s \in \mathbb{R}^{N_k \times d}$
 4: Compute attention: $\widetilde{P} = \mathrm{softmax}(Q_s K_s^\top / \sqrt{d})$
 5: MaxPool over $k \times k$ blocks to get $P_{\mathrm{imp}} \in \mathbb{R}^{N_b \times N_b}$
 6: **for** each row $i$ in $P_{\mathrm{imp}}$ **do**
 7: $\quad \tilde{P}_{\mathrm{imp}}(i,j) \leftarrow \dfrac{P_{\mathrm{imp}}(i,j)}{\sum_k P_{\mathrm{imp}}(i,k)}$
 8: $\quad$ Sort $\tilde{P}_{\mathrm{imp}}[i,:]$ descending $\rightarrow s$
 9: $\quad$ Find smallest $m$ such that $\sum_{j=1}^{m} s_j \geq \tau$, then clamp $m$ within the range defined by minimum
       and maximum retention ratios
10: $\quad$ Set $M[i,j] = 1$ for top $m$ indices, others $= 0$
11: **end for**
12: **return** Binary mask $M$

---

## D    MODEL CONFIGURATION DETAILS

Table 11 presents the detailed model configurations. For the distillation training phase (iterations 100-200), experiments were conducted on $8 \times$ NVIDIA A800 (80GB) GPUs using DeepSpeed ZeRO-2. The training took approximately 10 hours with a global batch size of 128, and the peak memory usage reached 76 GB.

Table 11: Detailed configuration parameters for Wan2.1-1.3B and CogVideoX-5B models.

| Category | Parameter | Wan2.1-1.3B | CogVideoX-5B |
|---|---|---|---|
| | Number of Layers | 30 | **42** |
| | Number of Attention Heads | 12 | **48** |
| | Attention Head Dimension | **128** | 64 |
| | In/Out Channels | 16 | 16 |
| | Temporal Compression Ratio | 4 | 4 |
| Model Architecture | Prediction Dtype | flow | velocity |
| | Sequence Length | **32760** | 17550 |
| | Text Dimension | 4096 | 4096 |
| | Patch Size | [1,2,2] | [1,2,2] |
| | Vocab Size | **256384** | 32128 |
| | Number of Timesteps | 1000 | 1000 |
| | Student learning rate | 1e-4 | 1e-4 |
| | Fake model learning rate | 5e-4 | 5e-4 |
| | LoRA Enabled | True | True |
| | LoRA alpha | 64 | 64 |
| | Optimizer | AdamW | AdamW |
| | Adam Beta1 | 0 | 0 |
| Training & Inference | Adam Beta2 | 0.95 | 0.95 |
| | Gradient Clipping | 1.0 | 1.0 |
| | Seed | 42 | 42 |
| | CFG | 5 | 6 |
| | Video Resolution | 480×832 | 480×720 |
| | Sample FPS | 16 | 8 |
| | Gradient Checkpointing | True | True |
| | Training Mode | Zero2 | Zero2 |

# E    RUNTIME BREAKDOWN OF THE ADAPTIVE BLOCK-SPARSE ATTENTION OPERATOR

We provide additional runtime analysis of our adaptive block-sparse attention (ASA) operator, including per-component breakdowns under two sequence lengths (100k and 18k), followed by a sparsity sweep comparing against FlashAttention2 (FA2) and FlashAttention3 (FA3). All results are measured with 30 heads and head dimension 128.

For clarity, the first row in each table reports the *end-to-end* forward time of ASA, while the remaining rows decompose this total into three main stages: (i) **Block Importance Estimation** (computing the importance scores for each KV block), (ii) **Mask Construction** (converting importance scores into a binary block-sparse mask), and (iii) **Block-Sparse Attention Compute** (the main attention kernel executed on the selected blocks), plus a small **Other** category for residual overheads (reshapes, indexing, etc.).

## E.1    COMPONENT-LEVEL BREAKDOWN

At sparsity $\tau = 0.8$, a simple FLOP-based model suggests that a block-sparse attention kernel should cost roughly $(1 - \tau)$ times the full-attention baseline (FA2), i.e., a $5\times$ theoretical speedup. In our setting, FA2 at 100k tokens takes $\approx 440\,\text{ms}$, so the ideal runtime under $0.8$ sparsity would be $\approx 88\,\text{ms}$. At 18k tokens, FA2 takes $13.16\,\text{ms}$, yielding an ideal $0.8$-sparse runtime of $\approx 2.63\,\text{ms}$.

**Runtime breakdown at 100k and 18k tokens (sparsity 0.8).**    Table 12 reports a component-level breakdown of the Adaptive Sparse Attention (ASA) operator at two sequence lengths under the same FLOP-equivalent sparsity of $0.8$.

Table 12: Runtime breakdown of Adaptive Sparse Attention (ASA) at two sequence lengths (sparsity 0.8).

|  | 100k tokens | | 18k tokens | |
| --- | --- | --- | --- | --- |
| **Component** | **Time (ms)** | **%** | **Time (ms)** | **%** |
| Adaptive Sparse Attention (Total) | 116.99 | 100.0% | 8.07 | 100.0% |
| Block Importance Estimation | 9.34 | 8.0% | 1.36 | 16.8% |
| Mask Construction | 1.03 | 0.9% | 0.41 | 5.1% |
| Block-Sparse Attention Compute | 106.32 | 90.9% | 6.06 | 75.1% |
| Other | 0.31 | 0.3% | 0.25 | 3.0% |

**100k tokens, sparsity 0.8.**    At this sequence length, runtime is dominated by the block-sparse compute kernel: over $90\%$ of the ASA time is spent in the block-sparse attention compute, while importance estimation and mask construction together account for only $\sim 9\%$ (Table 12). Compared to the FLOP-based ideal of $\approx 88\,\text{ms}$, the measured ASA runtime at 100k tokens is $116.99\,\text{ms}$, i.e., $1.33\times$ higher than the ideal. The resulting $\approx 29\,\text{ms}$ gap can be decomposed into $\approx 10.4\,\text{ms}$ from importance estimation + mask construction ($\sim 36\%$ of the gap, only $\sim 2.4\%$ of the dense FA2 time), and $\approx 18.6\,\text{ms}$ from the block-sparse compute kernel itself (padding, load imbalance, and memory/system overhead that are not captured by the simple FLOP count). In other words, at long sequences we recover about $3.8\times$ speedup over FA2 (vs. the $5\times$ theoretical limit), and the dominant source of the gap is the block-sparse kernel rather than mask building.

**18k tokens, sparsity 0.8.**    For shorter sequences, the relative share of preprocessing becomes more visible, and fixed kernel overheads dominate the deviation from the ideal FLOP-level speedup (Table 12). Here, the FLOP-based ideal runtime is $\approx 2.63\,\text{ms}$ (i.e., $0.2 \times 13.16\,\text{ms}$), whereas the measured ASA runtime is $8.07\,\text{ms}$. This corresponds to a $1.63\times$ speedup over FA2, recovering only $\sim 33\%$ of the theoretical $5\times$ limit. The $\approx 5.4\,\text{ms}$ gap to the ideal can again be decomposed into $\approx 1.8\,\text{ms}$ from importance estimation + mask construction ($\sim 33\%$ of the gap, $\sim 13.5\%$ of the dense FA2 time) and $\approx 3.7\,\text{ms}$ from the block-sparse compute kernel. Thus, in the short-sequence regime, the main discrepancy between theoretical and realized speedup comes from fixed kernel and

system overheads (e.g., launch latency, limited parallelism, and padding), while the cost of building the sparse mask remains small in absolute terms.

Overall, the gap between the FLOP-based ideal and the realized ASA runtime arises from both the mask-generation stage and the practical inefficiencies of the block-sparse compute kernel. In the short-sequence regime, the relative overhead of mask generation appears larger because several per-layer operations incur fixed costs that do not diminish proportionally with sequence length. As the sequence length increases, these fixed components become quickly amortized, causing the mask-generation share to shrink, and leaving the kerneltheoretical mismatch as the dominant contributor to the remaining gap.

# F VISUAL COMPARISON GALLERY

## F.1 ASA PERFORMANCE UNDER DIFFERENT SPARSITY LEVELS ON WAN2.1-1.3B

Each panel shows rows corresponding to different threshold values ($\tau \in \{0.5, 0.6, 0.8, 0.9\}$), with sparsity levels shown alongside. Lower thresholds exhibit severe structural distortion due to aggressive pruning, while higher thresholds restore coherent motion and fine-grained details. **Top:** results for prompt *"[A small boy, head bowed in determination, sprints through a torrential downpour as lightning crackles and thunder rumbles in the distance. Sheets of rain lash the ground, while the faint silhouette of a cozy home in the background glows like a small beacon of safety and warmth. ]"*. **Bottom:** results for prompt *"[A golden retriever in black sunglasses sprints across a rain-damp rooftop terrace, its long fur rippling in the breeze. Seen from a distance, the dog bounds toward the camera, tail wagging, as water droplets sparkle on the concrete and its golden coat stands out against the overcast sky. ]"*. The final row in each panel shows the dense-attention baseline.

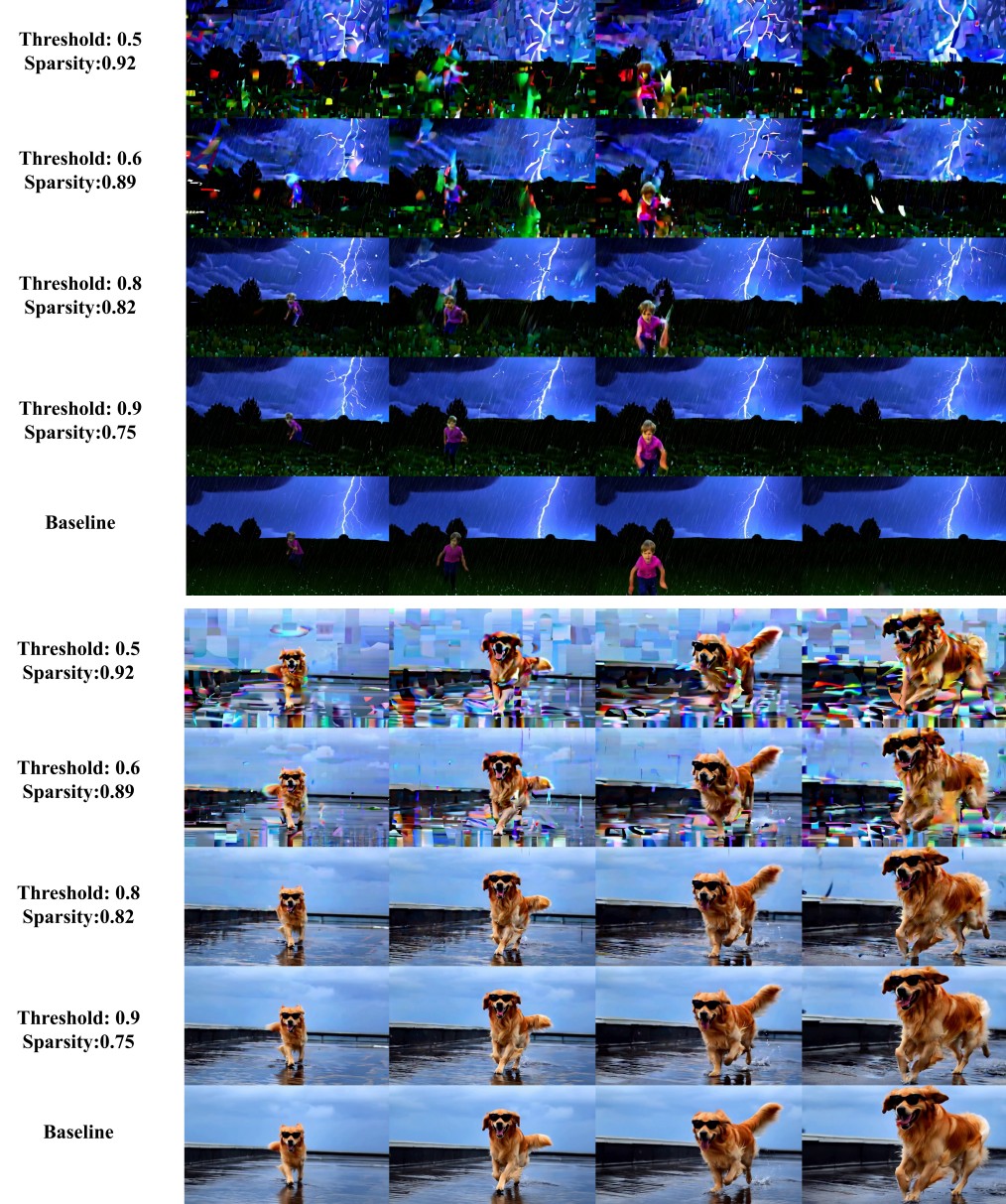

Figure 5: **Effect of attention threshold $\tau$ on visual quality across different prompts.**

## F.2 WAN2.1-1.3B RESULTS

Baseline

ASA_G

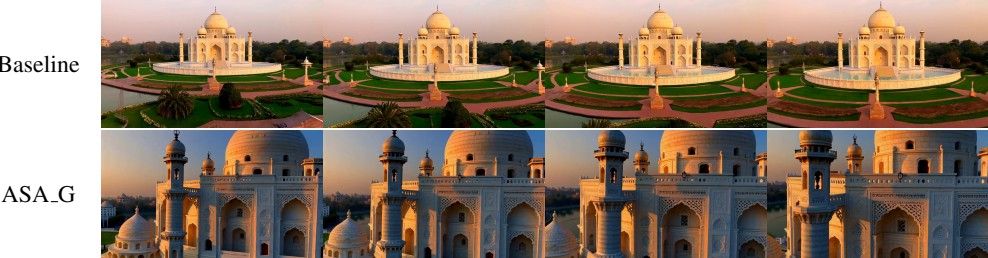

Figure 6: *"The camera orbits around. Taj Mahal, the camera circles around."*

Baseline

ASA_G

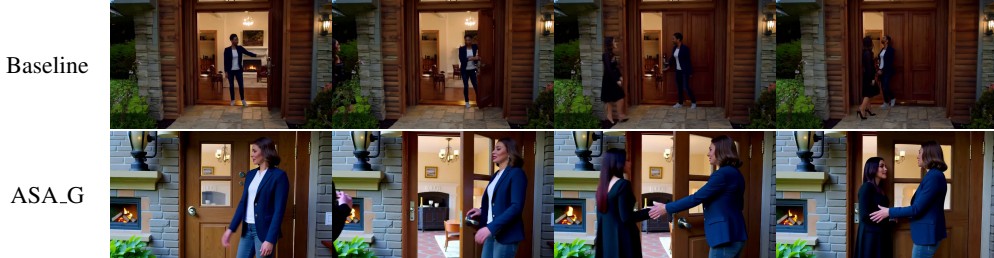

Figure 7: *"One person opens the door for another person."*

Baseline

ASA_G

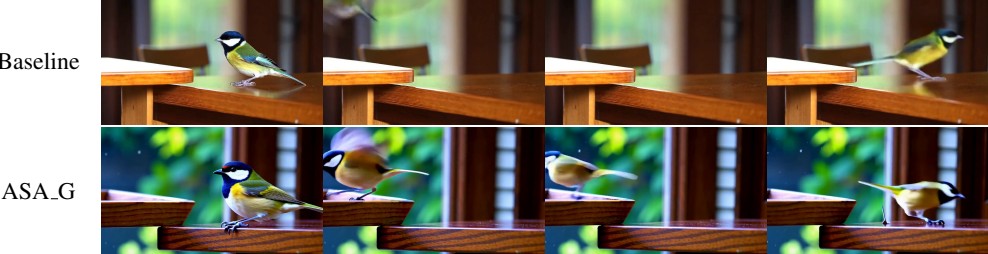

Figure 8: *"A bird is in front of a table, then the bird flies to the right of the table."*

Baseline

ASA_G

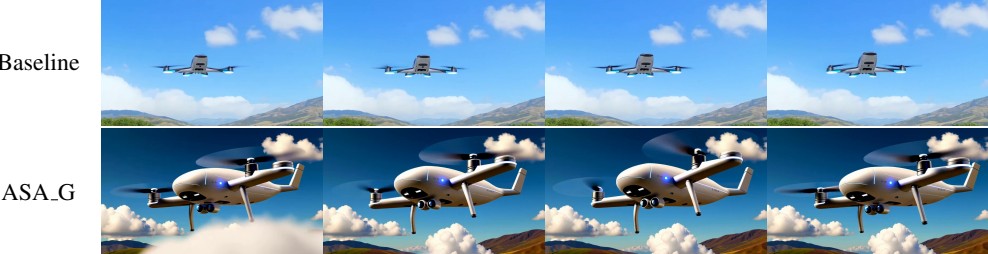

Figure 9: *"A drone is floating in the air."*

### F.3 COGVIDEOX-5B RESULTS

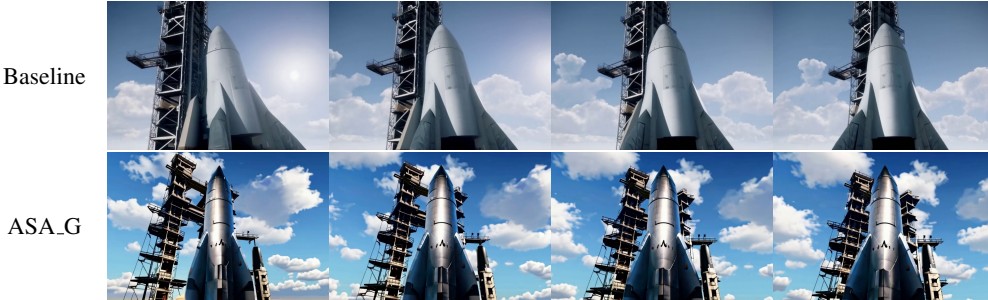

Figure 10: *"The camera orbits around. Rocket, the camera circles around."*

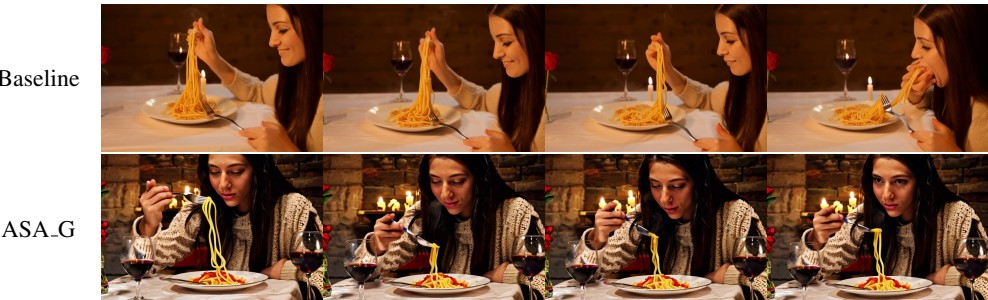

Figure 11: *"A person is eating spaghetti with a fork."*

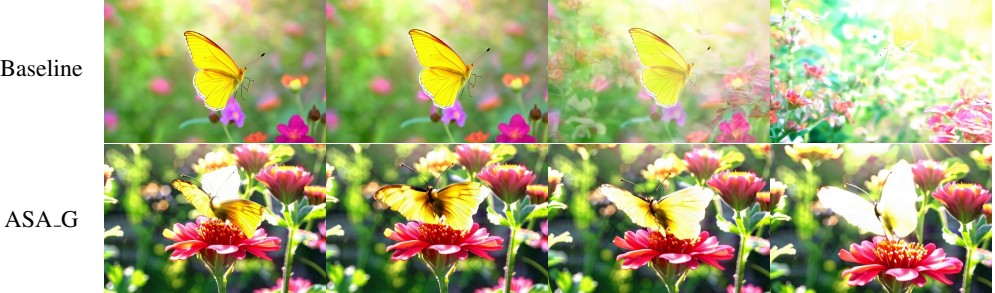

Figure 12: *"A butterfly's wings change from yellow to white."*

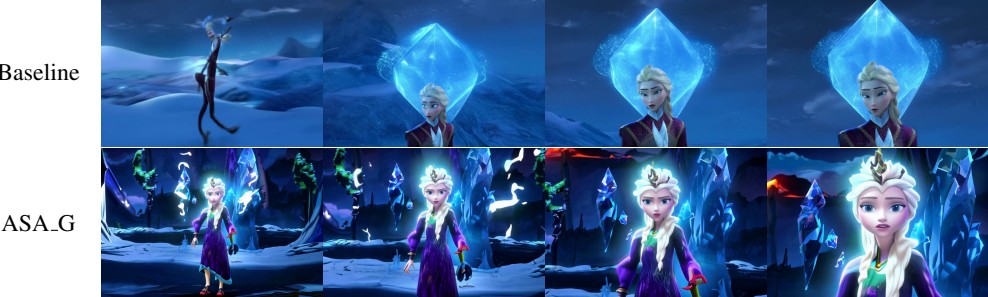

Figure 13: *"Princess Elsa plunged her northern kingdom into eternal winter."*

This section presents qualitative comparisons between baseline models and our ASA_G-distilled 8-step models. For each comparison, the top row shows results from the baseline 50-step model, while the bottom row shows results from our ASA_G method using only 8 steps(with sparsity ratios of 0.8 for Wan2.1-1.3B and 0.82 for CogVideoX-5B). Each row displays 4 sampled frames from the generated video sequence, demonstrating temporal consistency and visual quality across different prompts.

## G   ADVANTAGES OF JOINT TRAINING OVER TWO-STAGE PIPELINE

We compare our approach against a conventional two-stage pipeline that separates optimization into two independent tasks: Step Distillation then Sparse Fine-tuning. Our joint training strategy offers two critical advantages:

**1. Elimination of Dataset Bias.** A decoupled two-stage approach inherently relies on external real datasets for the sparse fine-tuning stage. This introduces significant dataset biasthe resulting model's quality becomes heavily dependent on the domain alignment and quality of the collected data. In contrast, BLADE is fully **data-free**. By jointly optimizing sparsity with distillation, we leverage the supervisory signals generated by the Teacher model as guidance. This ensures the student aligns perfectly with the teacher's distribution without introducing external dataset bias.

**2. Superior Training Efficiency.** As analyzed in **Appendix D**, our joint method achieves convergence within the same iteration budget (100–200 steps) as standard dense distillation. Since sparsity is applied *during* training, the computational cost per iteration is reduced compared to dense processing. Thus, BLADE achieves both acceleration targets (step reduction and sparsity injection) in a single, efficient pass, avoiding the substantial computational overhead of a separate sequential fine-tuning stage.

