# OpenReview forum: "BLADE: Block-Sparse Attention Meets Step Distillation for Efficient Video Generation"
_ICLR.cc/2026/Conference — ICLR 2026 Poster_

### Official Review · Reviewer_Sg3J · 2025-10-25

**Soundness:** 2
**Presentation:** 3
**Contribution:** 2
**Rating:** 4
**Confidence:** 4

**Summary:**

BLADE unifies sparse attention and step distillation to accelerate video diffusion transformers. It introduces Adaptive Block-Sparse Attention (ASA), which dynamically selects salient regions for computation, and integrates this sparsity directly into a Trajectory Distribution Matching (TDM) distillation process. This joint, data-free approach enables efficient few-step video generation without quality loss. Tested on CogVideoX-5B and Wan2.1-1.3B, BLADE achieves up to 14× faster inference and improved VBench-2.0 scores, showing that sparsity-aware distillation yields both speed and fidelity gains

**Strengths:**

1. Unifies two dominant acceleration axes—few-step distillation (via TDM) and sparse attention—by making sparsity part of the distillation loop rather than a post-hoc swap. The results matches current practice in few-step distillation (Trajectory Distribution Matching).

2. A diverse set up metric, including VBench, human evaluation, and PSNR/SSIM.

3. Offering both ASA (inference-only) and ASA-G (distillation-aware with global tokens) gives a practical path for immediate speedups and better quality when (light) training is allowed.

**Weaknesses:**

1. The paper discusses SpargeAttention and VSA but does not include either as baselines. I find ASA (inference-only) to highly resembles SpargeAttention, while VSA represents a trainable sparse attention design closely related to the ASA-G variant.  Moreover, the claim that VSA is limited by video resolution is inaccurate — VSA is not constrained by resolution in their open-sourced implementation. I feel including SpargeAtteniton as baseline is necessary (published at ICML) and including VSA is optional (a more recent work), but the author should discuss the difference.

2.  The distillation component essentially follows Trajectory Distribution Matching (TDM). The overall framework is thus a straightforward combination of two existing techniques. While I find incremental A+B paper to be acceptable, it should be supported by strong results. However,  this paper’s experimental evidence is weak, as discussed in later points.

3. One of the central arguement of this paper is combining sparse attention and distillation in a single training stage is better than doing them sequentially. However, there is not comparions against a. sparse attention tuning and then distillation. b. distillation and then sparse attention tuning.

4. The experimental evaluation is limited to small or medium-sized models and short video sequences at low resolution. No experiments are conducted on Hunyuan Video or Wan 2.1 14B, and no results are presented for long-sequence or high-resolution scenarios — where sparse attention truly matters due to quadratic scaling of attention cost.

5. The paper claims that ASA improves generation quality, but this is not supported by sufficient evidence. On Wan 1.3B, full attention after distillation achieves a higher VBench score. The only improvement shown is on CogVideoX, an arguably old model released last year, which does not strongly support the generality of the quality improvement claim.

6. In Table 2, the authors evaluate on the H20 GPU but use FlashAttention-2 (FA2) as the dense baseline. On H20, FlashAttention-3 (FA3) should be the standard baseline, as it is roughly 40% faster than FA2. Given the reported effective sparsity rate of 0.798, a theoretical 5× speedup over FA3 is expected, yet only a 3.3× gain over FA2 is reported — indicating significant implementation inefficiencies and optimization headroom that are not analyzed.

7. This claim  “Moreover, on models such as CogVideoX-5B with short video sequence lengths, our framework delivers a robust 8.89× speedup” is misleading in the abstract, the speedup mostly comes from distillation (7.93), putting this sentence make people think sparse attention play a huge part.

8. I believe DMD2 alone can reduce the number of inference steps to 3 or 4 steps on Wan 2.1, which is faster than the proposed sparse attention + TDM solution.

**Questions:**

See weakness section.

---

> ### Author Response · Authors · 2025-11-21
> **Response to Reviewer Sg3J (Part 1)**
>
> >## Weaknesses1&3
>
> **Table 8: Comparison of training-free sparse attention methods on Wan2.1-14B. ASA achieves the best trade-off between sparsity and visual quality.**
>
> | Method | Sparsity | PSNR $\uparrow$ | SSIM $\uparrow$ | LPIPS $\downarrow$ |
> | :--- | :---: | :---: | :---: | :---: |
> | SVG | 0.75 | 24.86 | 0.823 | 0.094 |
> | Sparge | 0.77 | 24.03 | 0.808 | 0.117 |
> | STA | 0.75 | 25.00 | 0.845 | 0.079 |
> | **ASA (Ours)** | 0.77 | **26.05** | **0.865** | **0.050** |
>
> **A1:**
> Thank you for your suggestion. As shown in Table 8, we benchmark it against state-of-the-art training-free sparse attention methods (SpargeAttention included) on **Wan2.1-14B**. ASA consistently outperforms existing training-free sparse attention baselines, achieving the best PSNR, SSIM, and the lowest LPIPS (0.050). These results mirror the trends observed on the 1.3B model and confirm that ASA’s dynamic masking strategy remains effective on larger architectures and longer sequences. More detailed analysis is provided in **Appendix A.5**.
>
> While ASA shares the high-level goal of inference-time sparsification with SpargeAttention, it differs in how block importance is estimated: ASA samples $k$ tokens per block, builds an attention map over all $(B \cdot k)$ sampled query and key tokens (where $B$ is the number of blocks), and applies max-pooling on each $k \times k$ sub-block to obtain block importance, preserving intra-block structure and yielding a more accurate estimate of block importance.
> In contrast, SpargeAttention compresses each block into a single mean token and derives importance from a $B \times B$ attention map, which provides a coarser approximation and can miss fine-grained block details.
>
> Regarding VSA, we agree it is closely related to our trainable variant (ASA-G); however, VSA fuses a block-sparse branch and a downsampled branch via learned gating vectors, whereas ASA-G instead extends the KV representations and fuses coarse and fine information directly within the attention computation, without introducing additional gating parameters.
>
> We appreciate your careful reading and thank you for pointing out the inaccurate wording regarding VSA. We have clarified in the paper that VSA is a dynamic, trainable sparse attention method operating on video latents as a 3D token grid. We now explicitly state that its limitation arises from the need to pad irregular latent shapes to tile-aligned sizes (e.g., padding CogVideoX-5B latents from $(45, 30, 13) \approx 17\text{k}$ tokens to $(48, 32, 16) \approx 24\text{k}$ tokens), which introduces overhead, rather than from intrinsic resolution constraints. We have updated the corresponding description in the Introduction section to reflect this correction.
>
> >## Weaknesses2
>
> **A2:** We have analyzed both potential pipelines and summarize the rationale for our approach below:
> * **Pipeline (a) [Sparse Tuning $\rightarrow$ Distillation]:** We appreciate the suggestion to explore Pipeline (a). However, our analysis suggests that if the model undergoes sparse tuning first, the following distillation stage necessitates sparsity constraints to prevent the dense distillation from reverting the learned patterns. Consequently, the second stage would function identically to our "Joint Training" method. Thus, while valid, this pipeline acts as "Sparse Pre-training followed by Joint Training", incurring additional computational costs compared to the simultaneous optimization in our framework.
> * **Pipeline (b) [Distillation $\rightarrow$ Sparse Finetuning]:** We compare our joint strategy against this sequential baseline where optimization is split into two separate tasks: **Step Distillation** and **Sparse Finetuning**. Our approach offers two distinct advantages:
>     * **Elimination of Dataset Bias:** The Sparse Finetuning phase typically requires external real datasets to recover performance. This introduces dataset bias, where the resulting quality depends heavily on the choice and domain of the external data. In contrast, our pipeline is fully **data-free**. By optimizing sparsity jointly with distillation, we leverage the teacher model's output as guidance, ensuring better alignment with the original distribution without introducing external bias.
>     * **Computational Efficiency:** The sequential pipeline requires running Dense Distillation and Sparse Finetuning separately. Our method achieves both objectives in a single pass. As shown in Appendix D, by incorporating sparsity *during* the distillation process, we accelerate individual training iterations while maintaining similar convergence speeds (100--200 iterations), effectively eliminating the computational cost of a separate fine-tuning stage.

---

> ### Author Response · Authors · 2025-11-21
> **Response to Reviewer Sg3J (Part 2)**
>
> >## Weaknesses4
>
> **A4:** We would like to clarify that we do not claim ASA itself improves generation quality; rather, it is the joint training pipeline that enhances the final performance.
>
> * *Clarification on Quality:* Our contribution centers on the efficacy of the Joint Training pipeline, rather than the inherent superiority of sparsity alone. Analogous to how DMD2 leverages distillation to enable high-quality generation at low step counts, our method utilizes joint distillation to adapt the model for high-quality inference under both low-step and sparse conditions. We do not imply that sparsity inherently outperforms dense attention; rather, we demonstrate that our pipeline effectively preserves high generation quality even under aggressive acceleration settings.
> * *On CogVideoX-5B:* We utilized this model specifically to validate architectural robustness. Despite its release date, it serves as a valuable benchmark to prove that our Joint Training strategy generalizes across different transformer architectures (delivering $8.89\times$ speedup), rather than being overfitted to a single model type.
>
> ---
>
> >## Weaknesses5
>
> **Table 12: Runtime breakdown of ASA at two sequence lengths (sparsity = 0.8).**
>
> | Component | 100k Time (ms) | 100k % | 18k Time (ms) | 18k % |
> | :--- | :---: | :---: | :---: | :---: |
> | Adaptive Sparse Attention (Total) | 116.99 | 100% | 8.07 | 100% |
> | Block Importance Estimation | 9.34 | 8.0% | 1.36 | 16.8% |
> | Mask Construction | 1.03 | 0.9% | 0.41 | 5.1% |
> | Block-Sparse Attention Compute | 106.32 | 90.9% | 6.06 | 75.1% |
> | Other | 0.31 | 0.3% | 0.25 | 3.0% |
> | FlashAttention2 (Dense Baseline) | 440.02 | - | 13.16 | - |
>
> **A5:**
> Thank you for your question.
> Below, we first analyze why the empirical speedup over the FA2 dense baseline falls short of the theoretical $5\times$ upper bound, and then clarify how ASA compares to FA3 and why FA2 is used as the primary dense baseline in the main paper.
>
> **1. Gap between the theoretical $5\times$ and observed speedup**
>
> The theoretical $5\times$ speedup assumes ideal FLOP reduction. However, as broken down in Table 12, practical performance is constrained by two sequence-dependent factors: Kernel Overhead (Dominant at 100k): For long sequences, the gap is primarily caused by the sparse kernel itself. Inherent overheads—such as mask loading, conditional checks, and thread load imbalance—prevent the compute from scaling perfectly linearly with FLOPs. Combined Overheads (Dominant at 18k): For shorter sequences, the costs of importance estimation and mask construction become significant relative to the total runtime and cannot be fully amortized. The limited speedup is thus a joint result of these unamortized mask costs and the inherent kernel inefficiencies mentioned above.
>
> **2. Comparison to FA3 and justification for using FA2 as the dense baseline**
>
> As you also point out, on Hopper-architecture GPUs, FA3 is typically about 40% faster than FA2, so FA3 is a natural dense baseline.
> Accordingly, we benchmarked ASA against this stronger baseline on H200. Under a typical high-sparsity configuration with 100k tokens and density $\approx 0.20$, ASA runs in **117.39 ms**, while FA3 takes **235.6 ms**, yielding roughly a **2$\times$** speedup.
> We do not achieve the ideal "$5\times$ over FA3" for two reasons: one comes from the same implementation factors that limit the speedup over FA2, and the other is that our sparse kernels have not yet integrated the additional low-level optimizations that make FA3 faster than FA2.
>
> Specifically, FA3 does not provide a Triton codebase that we can reuse, and Triton does not yet expose publicly documented fine-grained control over TMA on Hopper, making it challenging to faithfully reproduce FA3’s scheduling strategy within the Triton-based framework. Porting ASA to the official FA3 CUDA / CUTLASS kernels would require substantial engineering effort and a long development cycle, which is beyond the scope of this work at this stage.
> But we plan to integrate ASA with FA3’s Hopper optimized design (warp-specialized WGMMA, TMA-based asynchronous transfers, and GEMM--softmax overlap) to reach higher acceleration.
>
> We hope this clarifies both the source of the current gap and our choice of FA2 as the main dense baseline.

---

> ### Author Response · Authors · 2025-11-21
> **Response to Reviewer Sg3J (Part 3)**
>
> >## Weakness6
>
> **A6:** Our intention was not to attribute the entire speedup to sparse attention alone. Rather, we wanted to highlight the challenge of accelerating models with short sequence lengths (like CogVideoX-5B), where the $O(N^2)$ attention overhead is less dominant, making it difficult for sparse methods to provide significant additional gains. We intended to demonstrate that even in such challenging scenarios, our joint framework (combining step distillation and sparsity) can still push the performance boundary further than distillation alone. To prevent any misunderstanding, we have revised the sentence in the abstract to explicitly acknowledge the composite nature of this speedup:
>  -  "Notably, even on models like CogVideoX-5B with short video sequence lengths, where attention optimization is typically less impactful, our joint framework achieves a comprehensive $8.89\times$ speedup by effectively synergizing step distillation with sparse attention."
>
> ---
>
> >## Weakness7
>
> **Table 9: VBench-2.0 comparison of 4-step BLADE models against 50-step baselines. We report the end-to-end inference speedup relative to the dense 50-step baseline.**
>
> | Model | Method | Creativity | Commonsense | Control. | Human | Physics | Total | Speedup |
> | :--- | :--- | :---: | :---: | :---: | :---: | :---: | :---: | :---: |
> | CogVideoX-5B | Baseline | **0.458** | 0.523 | 0.341 | 0.808 | 0.539 | 0.534 | 1.0$\times$ |
> | | Ours | 0.425 | **0.553** | **0.389** | **0.840** | **0.606** | **0.562** | **15.2$\times$** |
> | Wan2.1-1.3B | Baseline | **0.508** | 0.549 | **0.338** | 0.820 | **0.600** | 0.563 | 1.0$\times$ |
> | | Ours | 0.467 | **0.564** | 0.333 | **0.897** | 0.594 | **0.570** | **17.6$\times$** |
>
> **A7:** Thank you for highlighting the potential of low-step inference. To investigate the feasibility of BLADE under strict computational constraints, we extend the distillation process to a lower-step regime of 4 sampling steps. Table 9 directly compares these 4-step sparse models against the standard 50-step dense baselines. Notably, our method achieves substantial end-to-end inference speedups of *15.2$\times$* on CogVideoX-5B and *17.6$\times$* on Wan2.1-1.3B. Despite this aggressive reduction in sampling steps, both models maintain superior VBench Total scores compared to the 50-step baselines (CogVideoX: 0.562 vs. 0.534; Wan2.1: 0.570 vs. 0.563). These results validate that our joint training framework remains highly effective even in few-step scenarios, ensuring high-quality generation alongside extreme acceleration. We have included these additional results and analysis in **Appendix A.6**.

---

> > ### Comment · Reviewer_Sg3J · 2025-11-27
> >
> > I appreatiate the authors' effort in the detailed rebuttal.  However, several of my original concerns remain insufficiently addressed:
> >
> > 1. The distinction between ASA’s sampling–max-pooling scheme and SpargeAttention’s mean-pooling is central to your method. Yet, why this component is not ablated in the original submission?
> >
> > 2. I still find this sentence to be too strong and a bit misleading: "Crucially, the acceleration is accompanied by a consistent quality improvement."
> >
> > 3. High resolution (720P) and better model (14B) experiments are still missing.
> >
> > 4. The efficiency gap remains substantial. For an acceleration method, I would expect the author to provide efficient implementation, and a theoretical 5× speedup translating to only ~2× in practice is difficult to justify.
> >
> > 5. Even with the revised wording, the dominant contributor to speedup in short-sequence models (e.g., CogVideoX-5B) remains step-distillation. The speedup should not be creditted to the distillation + sparse attention framework.
> >
> > I would thus keep my rating.

---

> ### Author Response · Authors · 2025-11-28
> **Rebuttal to the further questions of Reviewer Sg3J**
>
> **Q8:**
> > The distinction between ASA’s sampling–max-pooling scheme and SpargeAttention’s mean-pooling is central to your method. Yet, why this component is not ablated in the original submission?
>
> **A8:** We agree that this distinction is fundamental to our method's performance. While we did not include a specific internal ablation in the initial submission, we have addressed this in the revision through both theoretical clarification and empirical comparison:
>
> * **Theoretical Analysis (Revised Section 3.3):**  As elaborated in Step 1 of Section 3.3, unlike SpargeAttention, which collapses each block into a single mean token and derives importance from a coarse $N/b \times N/b$ attention map, ASA retains intra-block structure by computing attention over sampled tokens and then applying max-pooling within each sub-block. This finer-grained approximation enables ASA to better capture salient patterns within each block.
>
> * **Empirical Validation (Added in Appendix A.5):** To empirically validate this design, we included a direct comparison with SpargeAttention (which represents the mean-pooling approach) in Table 8. ASA significantly outperforms SpargeAttention (e.g., +2.02 PSNR and 50% lower LPIPS on Wan2.1-14B). This substantial performance gap serves as strong evidence that our sampling-max-pooling strategy provides a more accurate approximation of attention importance than the mean-pooling baseline.
>
> ---
>
> **Q9:**
> > I still find this sentence to be too strong and a bit misleading: "Crucially, the acceleration is accompanied by a consistent quality improvement."
>
> **A9:** We agree that the previous phrasing was imprecise. We have revised the sentence to explicitly state that the quality comparison is against the original 50-step baseline, ensuring the claim is accurate and not misleading.
>
> **Revised Text:**
> > "Crucially, the acceleration is achieved while maintaining generation quality comparable to the original 50-step baseline."
>
> ---
>
> **Q10:**
> > High resolution (720P) and better model (14B) experiments are still missing.
>
> **A10:**
>
> * **Sparse Attention Scalability (Refer to Table 8 in Response A1):** We have already provided the Wan2.1-14B evaluation in our previous update. As presented in Table 8 within Response A1, our method significantly outperforms other sparse attention baselines (e.g., +2.0 PSNR over SpargeAttention), validating the scalability of our sparse attention mechanism to large-scale models.
>
> * **Distillation Training Constraints:** We could not perform full distillation on 14B due to hardware limitations on our 8$\times$A800 (80GB) node. Holding three model copies requires $\approx$84GB for parameters alone, necessitating FSDP, sequence parallelism, and gradient accumulation. This results in an impractical $\approx$73 minutes per iteration. We will release the code with FSDP support to allow the community to reproduce this on H200-class clusters with superior compute power and larger memory.
>
> ---
>
> **Q11:**
> > The efficiency gap remains substantial. For an acceleration method, I would expect the author to provide efficient implementation, and a theoretical 5$\times$ speedup translating to only ~2$\times$ in practice is difficult to justify.
>
> **A11:**
> For Hopper-based GPUs, we implemented a BlockSparseAttention kernel with the FA3 backend using FlashInfer (with minor modifications to VariableBlockSparseAttentionWrapper for faster KV loading; code is updated in the supplementary material).
>
> On 100k-token sequences at 0.8 sparsity (dense ratio = 0.20), ASA runtime is 74.27 ms, compared to 234.42 ms for FA3, achieving **$\approx$3.15$\times$ speedup**. Since this wrapper uses variable-sized blocks, its pipeline efficiency, memory access regularity, and register/shared-memory usage are still inferior to a block sparse attention kernel with fixed block size, leaving some gap to the theoretical upper bound. We plan to develop a FA3 backend block sparse attention kernel with fixed block size to further reduce this gap.
>
> On non-Hopper GPUs (e.g., A100, seqlen 100k), our original implementation runs in 212.06 ms versus 836.57 ms for FA2 at 20% density, yielding a **3.94$\times$ speedup** close to the theoretical limit.
>
> ---
>
> **Q12:**
> > Even with the revised wording, the dominant contributor to speedup in short-sequence models (e.g., CogVideoX-5B) remains step-distillation. The speedup should not be credited to the distillation + sparse attention framework.
>
> **A12:** We agree that the speedup on short-sequence models is dominated by step distillation. We have removed the claim regarding "synergizing" or "joint framework" benefits for CogVideoX-5B in the abstract. The revised text now objectively reports the speedup figures without attributing them to sparse attention optimizations.
>
> **Revised Abstract:**
> > "On Wan2.1-1.3B, BLADE achieves a $14.10\times$ end-to-end inference acceleration over a 50-step baseline, and an $8.89\times$ speedup on the short-sequence model CogVideoX-5B."

---

### Official Review · Reviewer_WLSr · 2025-10-27

**Soundness:** 3
**Presentation:** 3
**Contribution:** 3
**Rating:** 6
**Confidence:** 4

**Summary:**

The paper proposes BLADE, which unifies Adaptive Block-Sparse Attention (ASA) and Sparsity-Aware Step Distillation, introducing a new sparse attention operator to enhance efficiency. The method achieves 8.9×–14.1× acceleration on CogVideoX-5B and Wan2.1-1.3B with maintained or improved quality. The idea is clear, technically solid, and promising for efficient video generation, though validation on larger models (e.g., Wan2.1-14B) is still needed.

**Strengths:**

- Effective under both training-free and distillation-based settings.

- Large speedups with stable quality (VBench and human evaluation confirmed).

- Robust at high sparsity (~80%), outperforming similar methods.

- Detailed pseudocode and source code are provided, making the method easy to follow and reproduce.

**Weaknesses:**

- Lacks large-scale and long-sequence experiments;

- ASA is currently implemented in a custom Triton kernel and Block Sparse Attention library, and a more detailed analysis of the runtime contribution of each component would be helpful.

**Questions:**

Have you considered including training-free results on larger models such as Wan2.1-14B to strengthen the evaluation?

Could you provide a more detailed breakdown of the runtime for each component of ASA to better explain the performance gap?

---

> ### Author Response · Authors · 2025-11-21
> **Response to Reviewer WLSr**
>
> **Q1:** Have you considered including training-free results on larger models such as Wan2.1-14B to strengthen the evaluation?
>
> **A1:** **Table 8:** Comparison of training-free sparse attention methods on Wan2.1-14B. ASA achieves the best trade-off between sparsity and visual quality.
>
> | Method | Sparsity | PSNR $\uparrow$ | SSIM $\uparrow$ | LPIPS $\downarrow$ |
> | :--- | :---: | :---: | :---: | :---: |
> | SVG | 0.75 | 24.86 | 0.823 | 0.094 |
> | Sparge | 0.77 | 24.03 | 0.808 | 0.117 |
> | STA | 0.75 | 25.00 | 0.845 | 0.079 |
> | **ASA (Ours)** | 0.77 | **26.05** | **0.865** | **0.050** |
>
> Thank you for your suggestion. To validate the scalability of ASA to larger architectures and longer video sequences, we benchmark it against state-of-the-art training-free sparse attention methods on Wan2.1-14B. As shown in **Table 8**, ASA maintains superior performance at comparable sparsity levels. Notably, our method achieves an exceptionally low LPIPS of 0.050, outperforming the strongest baseline, STA, by a significant margin (0.029 improvement). Compared to SpargeAttention, ASA improves PSNR by over 2.0 and reduces perceptual loss by more than 50%, underscoring its capability to preserve fine-grained semantic details even when processing extended contexts. These results, consistent with the 1.3B evaluation, demonstrate that ASA's dynamic masking strategy is robust across both model scales and sequence lengths.
>
> ---
> **Q2:** Could you provide a more detailed breakdown of the runtime for each component of ASA to better explain the performance gap?
>
> **A2:** **Table 12:** Runtime breakdown of ASA at two sequence lengths (sparsity = 0.8).
>
> | Component | 100k tokens (ms) | 100k tokens (%) | 18k tokens (ms) | 18k tokens (%) |
> | :--- | :---: | :---: | :---: | :---: |
> | Adaptive Sparse Attention (Total) | 116.99 | 100% | 8.07 | 100% |
> | Block Importance Estimation | 9.34 | 8.0% | 1.36 | 16.8% |
> | Mask Construction | 1.03 | 0.9% | 0.41 | 5.1% |
> | Block-Sparse Attention Compute | 106.32 | 90.9% | 6.06 | 75.1% |
> | Other | 0.31 | 0.3% | 0.25 | 3.0% |
> | FlashAttention2 (Dense Baseline) | 440.02 | - | 13.16 | - |
>
> Thank you for your question. At sparsity $\tau=0.8$, a simple FLOP-based model suggests that a block-sparse attention kernel should cost roughly $(1-\tau)$ times the full-attention baseline, i.e., a $5\times$ theoretical speedup. In our setting, FA2 at 100k tokens takes $\approx 440$ ms, so the ideal runtime under $0.8$ sparsity would be $\approx 88$ ms. At 18k tokens, FA2 takes $13.16$ ms, yielding an ideal $0.8$-sparse runtime of $\approx 2.63$ ms.
>
> **100k tokens, sparsity 0.8.**
> At this sequence length, runtime is dominated by the block-sparse compute kernel, and the cost of importance estimation and mask construction is small in both absolute and relative terms.
>
> Compared to the FLOP-based ideal of $\approx 88$ ms, the measured ASA runtime at 100k tokens is 116.99 ms, i.e., $1.33\times$ higher than the ideal. The resulting $\approx 29$ ms gap can be decomposed into $\approx 10.4$ ms from importance estimation + mask construction ($\sim$ 36% of the gap, only $\sim$ 2.4% of the dense FA2 time), and $\approx 18.6$ ms from the block-sparse compute kernel itself (padding, load imbalance, and memory/system overhead that are not captured by the simple FLOP count). As a result, we recover about $3.8\times$ speedup over FA2 (vs. the $5\times$ theoretical limit).
>
> **18k tokens, sparsity 0.8.**
> For shorter sequences, the relative share of preprocessing becomes more visible, and fixed kernel overheads dominate the deviation from the ideal FLOP-level speedup.
>
> Here, the FLOP-based ideal runtime is $\approx 2.63$ ms (i.e., $0.2 \times 13.16$ ms), whereas the measured ASA runtime is 8.07 ms. This corresponds to a $1.63\times$ speedup over FA2, recovering only $\sim$ 33\% of the theoretical $5\times$ limit. The $\approx 5.4$ ms gap to the ideal can again be decomposed into $\approx 1.8$ ms from importance estimation + mask construction ($\sim$ 33% of the gap, $\sim$ 13.5% of the dense FA2 time) and $\approx 3.7$ ms from the block-sparse compute kernel. This shows that in the short-sequence regime, the main discrepancy between theoretical and realized speedup comes from fixed kernel and system overheads (e.g., launch latency, limited parallelism, and padding), while the cost of building the sparse mask remains small in absolute terms.
>
> Overall, the gap between the FLOP-based ideal and the realized ASA runtime arises from both the mask-generation stage and the practical inefficiencies of the block-sparse compute kernel. In the short-sequence regime, the relative overhead of mask generation appears larger because several per-layer operations incur fixed costs that do not diminish proportionally with sequence length. As the sequence length increases, these fixed components become quickly amortized, causing the mask-generation share to shrink, and leaving the kernel–theoretical mismatch as the dominant contributor to the remaining gap.

---

> > ### Comment · Reviewer_WLSr · 2025-11-27
> >
> > Thank you for your rebuttal.
> >
> > My concerns have been addressed. The newly added analysis and results are clear and significantly enhance the contribution of the paper. Therefore, I raise my score to 8 and recommend acceptance.

---

### Official Review · Reviewer_asWM · 2025-10-30

**Soundness:** 3
**Presentation:** 3
**Contribution:** 3
**Rating:** 6
**Confidence:** 4

**Summary:**

The paper introduces BLADE, a framework that integrates Adaptive Block-Sparse Attention (ASA) with step distillation for efficient video generation. It proposes a data-free joint training approach, leveraging ASA to generate dynamic, content-aware sparsity masks and sparsity-aware Trajectory Distribution Matching (TDM) to enhance quality. Experiments on CogVideoX-5B and Wan2.1-1.3B demonstrate significant speedups (up to 14.10×) and quality improvements, validated by VBench-2.0 and human evaluations.

**Strengths:**

- Integration of adaptive block-sparse attention with step distillation, enabling data-free joint training for efficient video generation.
- ASA mechanism dynamically generates content-aware sparsity masks that enable high sparsity levels, achieving hardware-friendly acceleration without quality loss when combined with distillation training.
- Demonstrates substantial speedups (up to 14.10×) on diverse models like CogVideoX-5B and Wan2.1-1.3B, with consistent quality improvements on VBench.

**Weaknesses:**

This paper lacks details on experimental settings and comparative results, for example:
- Lack of reporting on specific GPU hours, training batch size, and memory usage for the 100-200 distillation iterations.
- Lack of inference results demonstrating video quality across low-to-high sparsity levels to illustrate the impact.

**Questions:**

Please refer to the **Weaknesses** above.

---

> ### Author Response · Authors · 2025-11-21
> **Response to Reviewer asWM**
>
> **Q1: Lack of reporting on specific GPU hours, training batch size, and memory usage for the 100-200 distillation iterations.**
>
> **A1:** Thank you for the suggestion. For the distillation training phase (iterations 100-200), experiments were conducted on 8 × NVIDIA A800 (80GB) GPUs using DeepSpeed ZeRO-2. The training took approximately 10 hours with a global batch size of 128, and the peak memory usage reached 76 GB. We have included these detailed training configurations in **Appendix D**.
>
> ---
>
> **Q2: Lack of inference results demonstrating video quality across low-to-high sparsity levels to illustrate the impact.**
>
> **Table 7:** Ablation study on the attention threshold (τ) in ASA. Experiments are conducted on Wan2.1-1.3B (50 steps; ASA enabled after a 12-step warm-up). τ governs the sparsity-quality frontier.
>
> | Threshold (τ) | Sparsity | PSNR ↑ | SSIM ↑ | LPIPS ↓ |
> | :---: | :---: | :---: | :---: | :---: |
> | 0.9 | 0.73 | **23.93** | **0.856** | **0.106** |
> | 0.8 | 0.80 | 22.08 | 0.812 | 0.153 |
> | 0.7 | 0.84 | 20.46 | 0.750 | 0.209 |
> | 0.6 | 0.87 | 19.20 | 0.712 | 0.243 |
> | 0.5 | 0.92 | 15.48 | 0.583 | 0.436 |
>
> **A2:** We appreciate the suggestion to illustrate the impact of varying sparsity levels. The threshold τ directly modulates the trade-off between computational sparsity and generation fidelity. By varying τ from 0.5 to 0.9, we observe in Table 7 that higher thresholds preserve more attention blocks, naturally leading to superior quality metrics. A threshold of τ = 0.9 yields near-dense quality (PSNR 23.93) but results in a lower sparsity of 0.73. Conversely, aggressive pruning with τ = 0.5 achieves high sparsity (0.92) but causes a structural collapse in quality (PSNR drops to 15.48), indicating that critical attention contexts are being discarded.
>
> We identify τ = 0.8 as the optimal operating point, achieving a sparsity of 0.80 without severe degradation in perceptual quality (SSIM > 0.8). This ablation further shows that ASA supports flexible deployment: higher thresholds are suitable for quality-critical applications, while lower thresholds benefit latency-sensitive scenarios where minor artifacts are acceptable. Visual examples across different sparsity levels are provided in **Appendix F.1**.

---

### Official Review · Reviewer_YBVV · 2025-10-31

**Soundness:** 3
**Presentation:** 3
**Contribution:** 3
**Rating:** 6
**Confidence:** 3

**Summary:**

This paper presents BLADE, a novel framework designed to accelerate video diffusion models. BLADE significantly improves inference speed while maintaining or even enhancing generation quality by combining the dynamic, content-aware Adaptive Block-Sparse Attention (ASA) mechanism with the data-free Trajectory Distribution Matching (TDM) distillation process. Extensive experiments on various video models demonstrate significant improvements in kernel-level efficiency, end-to-end inference speed, and generation quality.

**Strengths:**

The innovative BLADE framework effectively addresses the computational bottleneck in accelerating inference for video diffusion models by jointly training the sparse attention mechanism (ASA) with trajectory distillation (TDM). This solution not only accelerates the generation process but also maintains high-quality outputs, especially in high sparsity conditions, achieving high-quality video generation with fewer steps, outperforming traditional methods. The paper is clearly motivated, well-written, and the diagrams are easy to understand, ensuring good readability.

**Weaknesses:**

1. Although ASA's performance is compared with traditional sparse attention methods (e.g., STA, RaA, SVG), the paper does not delve into the impact of different sparsity patterns (e.g., varying threshold settings, block sizes) on generation quality. Ablation experiments with different sparse configurations could provide further insights.
2. BLADE optimizes generation performance by jointly training sparse attention and trajectory distribution matching. The core innovation here is the fusion of sparsity with the distillation process. However, there is a lack of further experimental evidence to demonstrate the advantages of joint training.

**Questions:**

1. The paper mentions that ASA performs excellently in accelerating the generation process, but increasing sparsity might negatively impact generation quality. Specifically, how can the generation quality be ensured when sparsity is very high, while still maintaining significant computational acceleration?

---

> ### Author Response · Authors · 2025-11-21
> **Response to Reviewer YBVV (Part 1)**
>
> **Q1:**
> Although ASA's performance is compared with traditional sparse attention methods (e.g., STA, RaA, SVG), the paper does not delve into the impact of different sparsity patterns (e.g., varying threshold settings, block sizes) on generation quality. Ablation experiments with different sparse configurations could provide further insights.
>
> **Table 6: Ablation study on block size configuration for ASA on Wan2.1-1.3B (sparsity ratio 0.8). Smaller block sizes offer finer granularity but incur higher overhead.**
>
> | Block Size ($Q \times K$) | PSNR $\uparrow$ | SSIM $\uparrow$ | LPIPS $\downarrow$ |
> | :--- | :---: | :---: | :---: |
> | $64 \times 64$ | **22.24** | **0.818** | **0.144** |
> | $128 \times 64$ | 22.05 | 0.803 | 0.162 |
> | $128 \times 128$ | 21.75 | 0.793 | 0.169 |
>
> **Table 7: Ablation study on the attention threshold ($\tau$) in ASA on Wan2.1-1.3B. $\tau$ governs the sparsity-quality frontier.**
>
> | Threshold ($\tau$) | Sparsity | PSNR $\uparrow$ | SSIM $\uparrow$ | LPIPS $\downarrow$ |
> | :---: | :---: | :---: | :---: | :---: |
> | 0.9 | 0.73 | **23.93** | **0.856** | **0.106** |
> | 0.8 | 0.80 | 22.08 | 0.812 | 0.153 |
> | 0.7 | 0.84 | 20.46 | 0.750 | 0.209 |
> | 0.6 | 0.87 | 19.20 | 0.712 | 0.243 |
> | 0.5 | 0.92 | 15.48 | 0.583 | 0.436 |
>
> **A1:** We appreciate the suggestion to analyze different sparsity patterns. Block size determines the granularity of the attention masking, acting as a pivotal hyperparameter in the ASA mechanism. We analyze three block configurations under a constant sparsity ratio of 0.8. Table 6 demonstrates a monotonic improvement in generation quality as block size decreases. Specifically, the $64 \times 64$ configuration outperforms the coarser $128 \times 128$ setting by 0.49 in PSNR and 0.025 in SSIM.
>
> This performance gain stems from the finer granularity of smaller blocks, which allows the ASA mechanism to preserve salient regions with higher precision, adapting more effectively to local semantic structures. However, finer blocking introduces non-negligible computational overhead during mask generation and memory access. Consequently, we adopt $128 \times 128$ in our main experiments as a strategic trade-off, prioritizing system throughput while maintaining competitive generation quality.
>
> The threshold $\tau$ directly modulates the trade-off between computational sparsity and generation fidelity. By varying $\tau$ from 0.5 to 0.9, we observe in Table 7 that higher thresholds preserve more attention blocks, naturally leading to superior quality metrics. A threshold of $\tau=0.9$ yields near-dense quality but results in a lower sparsity of 0.73. Conversely, aggressive pruning with $\tau=0.5$ achieves high sparsity but causes a structural collapse in quality, indicating that critical attention contexts are being discarded.
>
> We identify $\tau=0.8$ as the optimal operating point, achieving a sparsity of 0.80 without severe degradation in perceptual quality (SSIM $>0.8$). This ablation further shows that ASA supports flexible deployment: higher thresholds are suitable for quality-critical applications, while lower thresholds benefit latency-sensitive scenarios where minor artifacts are acceptable. Visual examples across different sparsity levels are provided in **Appendix F.1**. Detailed analyses have been added to **Appendix A.3** and **A.4**.
>
> ---
>
> **Q2:**
> The paper mentions that ASA performs excellently in accelerating the generation process, but increasing sparsity might negatively impact generation quality. Specifically, how can the generation quality be ensured when sparsity is very high, while still maintaining significant computational acceleration?
>
> **A2:**
> While high sparsity inherently reduces information, our framework ensures high generation quality through two core mechanisms:
>
> * **Joint Training with Distillation:** Unlike standard post-training pruning methods that degrade quality when sparsity increases, our method incorporates sparsity *during* the training phase. By jointly optimizing the sparse student model with supervision from a dense teacher model (via step distillation), the student explicitly learns to reconstruct high-fidelity features using only the selected sparse tokens. This allows the model to adapt to the information loss and recover quality, effectively shifting the trade-off curve.
> * **Adaptive Importance-based Sampling:**
>     Rather than pruning blocks randomly, ASA estimates the importance of each KV block *for every query block* based on its attention scores, and then applies a threshold-based rule to retain only the most salient KV regions. In this way, even at high sparsity (e.g., 80%), computation is concentrated on semantically critical blocks, which substantially reduces information loss and helps preserve generation quality.
>
> As shown in Table 7, our method effectively maintains high structural integrity even at a very high sparsity level of 0.80. Detailed results and visualizations are provided in **Appendix A.4** and **F.1**.

---

> ### Author Response · Authors · 2025-11-21
> **Response to Reviewer YBVV (Part 2)**
>
> **Q3:**
> BLADE optimizes generation performance by jointly training sparse attention and trajectory distribution matching. The core innovation here is the fusion of sparsity with the distillation process. However, there is a lack of further experimental evidence to demonstrate the advantages of joint training.
>
> **A3:** We compare our approach against a conventional two-stage pipeline that separates optimization into two independent tasks: Step Distillation then Sparse Finetuning. Our joint training strategy offers two critical advantages:
>
> * **Elimination of Dataset Bias:** A decoupled two-stage approach inherently relies on external real datasets for the sparse finetuning stage. This introduces significant dataset bias—the resulting model's quality becomes heavily dependent on the domain alignment and quality of the collected data, potentially drifting from the original model's distribution. In contrast, BLADE is fully **data-free**. By jointly optimizing sparsity with distillation, we leverage the "synthetic" signals generated by the Teacher model as guidance. This ensures the student aligns perfectly with the teacher's distribution without introducing external dataset bias.
>
> * **Superior Training Efficiency:** As analyzed in **Appendix D**, our joint method achieves convergence within the same iteration budget (100--200 steps) as standard dense distillation. Since sparsity is applied *during* training, the computational cost per iteration is reduced compared to dense processing. Thus, BLADE achieves both acceleration targets (step reduction and sparsity injection) in a single, efficient pass, avoiding the substantial computational overhead of a separate finetuning stage.

---

### Author Response · Authors · 2025-12-01
**Rebuttal Summary by the Authors**

Dear Area Chair,

We sincerely appreciate the time and effort you have dedicated to reviewing our submission. We are encouraged by the positive feedback highlighting that our method is “clear, technically solid, and promising” (Reviewer **WLSr**), “effectively addresses the computational bottleneck” (Reviewer **YBVV**), achieves “hardware-friendly acceleration without quality loss” (Reviewer **asWM**), and “unifies two dominant acceleration axes” (Reviewer **Sg3J**).

**Update: Score Upgrade from Reviewer WLSr.** Reviewer WLSr raised their score from **6 $\to$ 8**, confirming that our rebuttal has fully addressed their concerns.

Below we summarize the main clarifications and additions to facilitate your assessment.

**I. Response to Shared Concerns**

Several reviewers raised similar constructive questions regarding scalability, efficiency, ablation details, and training rationale. We addressed these shared concerns through extensive new experiments and analysis:

* **Scalability on 14B Model** (Raised by *Sg3J, WLSr*):
    We benchmarked BLADE against state-of-the-art training-free methods (including *SpargeAttention* and *SVG*) on the **Wan2.1-14B** model.

    $\hookrightarrow$ **Result:** BLADE achieves the highest PSNR (**26.05** vs. 24.03 for Sparge) and lowest LPIPS (**0.050**), validating performance on large-scale backbones (**Table 8**).

* **Efficiency & Implementation** (Raised by *Sg3J, WLSr*):
    To address the gap between theoretical and practical speedup for Hopper-architecture GPUs, we upgraded our implementation to a **FlashAttention-3 (Hopper-optimized) backend**.

    $\hookrightarrow$ **Result:** This improved speedups to **3.15$\times$** on H200 and **3.94$\times$** on A100 (near theoretical limits,sparsity = 0.8). We also provided a detailed runtime breakdown for each ASA component in **Table 12**.

* **Ablation Studies on Sparsity** (Raised by *YBVV, asWM*):
    We conducted detailed ablations to quantify the trade-off between speed and quality.

    $\hookrightarrow$ **Result:** Added analysis on Block Size configurations (**Table 6**) and Sparsity Thresholds $\tau$ (**Table 7**), along with visual comparisons in **Appendix F**, demonstrating robust quality even at high sparsity.

* **Joint Training Efficiency & Rationale** (Raised by *YBVV, Sg3J*):
    We clarified that our joint training is significantly more efficient than sequential pipelines (Distillation $\to$ Finetuning). It achieves convergence within the same iteration budget while simultaneously eliminating dataset bias, avoiding the overhead of a separate finetuning stage (**Appendix G**).

**II. Response to Reviewer-Specific Feedback**

Beyond the shared concerns, we addressed the individual inquiries of each reviewer:

* **Reviewer Sg3J**
    * **Few-Step Inference:** Validated efficacy in a strict **4-step regime**, achieving **15$\times$--17$\times$** speedup with superior VBench scores (**Table 9**).
    * **Scientific Rigor:** Refined the **Abstract** to attribute short-sequence speedups to distillation, corrected the description regarding VSA's limitation in the **Introduction**, and clarified the distinction between our sampling-max-pooling strategy and baselines in **Section 3.3**.

* **Reviewer asWM**
    * **Training Cost:** Reported specific training metrics for the distillation phase in **Appendix D**: approx. 80 GPU hours, batch size 128, and 76GB peak memory.

We hope this summary assists in your assessment of our work. Thank you for your time and consideration.

Best regards,

The Authors

---

### Meta-Review · Area_Chair_g6sv · 2026-01-02

**Summary:**

Reviewers for the most part agree that this paper presents BLADE, a clear technically solid framework that unifies Adaptive Block-Sparse Attention (ASA) with sparsity-aware step distillation to accelerate video diffusion models. The idea of integrating sparsity directly into the distillation process rather than applying it post-hoc is viewed as intuitive and promising. Experiments also show substantial inference acceleration while maintaining generation quality comparable to a 50-step baseline.

**Reviewer Concerns:**

The main concerns focused on evidence for the benefit of joint training vs. sequential pipelines, evaluation at larger scales (longer sequences, higher resolution, larger models), and the gap between theoretical and realized sparse-attention speedups.
- These issues were largely addressed in the rebuttal through added training-free ASA evaluation on Wan2.1-14B, results for sparse attention, direct baseline comparisons, detailed runtime breakdowns, clarified wording around quality and speedup attribution, and explicit reporting of training cost and efficiency. Remaining concerns mainly relate to the absence of full distillation at very large scale and kernel-level optimization headroom.

**Reviewer Scores:**

Initial scores ranged from 6 (marginally above threshold) to 4 (marginally below). After rebuttal, WLSr raised their score to 8, and recommended accept. Reviewers YBVV and asWM remained positive, while Sg3J maintained their original rating. Overall, post-discussion support acceptance.

---

### Decision · Program_Chairs · 2026-01-26

Accept (Poster)